# The choroid plexus is an important circadian clock component

Jihwan Myung [1,2,3,4,5], Christoph Schmal[6], Sungho Hong [2], Yoshiaki Tsukizawa[7], Pia Rose[6], Yong Zhang[8], Michael J. Holtzman [8], Erik De Schutter [2], Hanspeter Herzel[6], Grigory Bordyugov[6] & Toru Takumi[1,7]

Mammalian circadian clocks have a hierarchical organization, governed by the suprachiasmatic nucleus (SCN) in the hypothalamus. The brain itself contains multiple loci that maintain autonomous circadian rhythmicity, but the contribution of the non-SCN clocks to this hierarchy remains unclear. We examine circadian oscillations of clock gene expression in various brain loci and discovered that in mouse, robust, higher amplitude, relatively faster oscillations occur in the choroid plexus (CP) compared to the SCN. Our computational analysis and modeling show that the CP achieves these properties by synchronization of "twist" circadian oscillators via gap-junctional connections. Using an in vitro tissue coculture model and in vivo targeted deletion of the *Bmal1* gene to silence the CP circadian clock, we demonstrate that the CP clock adjusts the SCN clock likely via circulation of cerebrospinal fluid, thus finely tuning behavioral circadian rhythms.

[1] RIKEN Brain Science Institute (BSI), Wako 351-0198, Japan. [2] Computational Neuroscience Unit, Okinawa Institute of Science and Technology, Okinawa 904-0495, Japan. [3] Graduate Institute of Humanities in Medicine, Taipei Medical University, Taipei 11031, Taiwan. [4] TMU-Research Center of Brain and Consciousness, Taipei Medical University, Taipei 11031, Taiwan. [5] Laboratory of Braintime, Shuang Ho Hospital, New Taipei City 23561, Taiwan. [6] Institute for Theoretical Biology, Charité-Universitätsmedizin and Humboldt Universität, Berlin D-10115, Germany. [7] Department of Anatomy, Hiroshima University School of Medicine, Hiroshima 734-8551, Japan. [8] Pulmonary and Clinical Care Medicine, Washington University School of Medicine, St. Louis, MO 63110, USA. These authors contributed equally: Jihwan Myung, Christoph Schmal, Sungho Hong. Correspondence and requests for materials should be addressed to J.M. (email: jhmyung@gmail.com) or to H.H. (email: h.herzel@biologie.hu-berlin.de) or to T.T. (email: toru.takumi@riken.jp)

The suprachiasmatic nucleus (SCN) is the gatekeeper to the circadian rhythms in the body. Known also as the master circadian clock, it keeps the bodily rhythms in sync with the light–dark cycle in the outside environment. The SCN is also an encoder of seasonal rhythms through phase reorganization among its subregions, in adaptation to daylength[1–3]. It propagates information of these external cycles to a web of internal circadian cycles in peripheral circadian clocks[4]. Circadian signaling is deemed hierarchical because of the assumed one-way flow of information from the SCN. However, the peak phases of the circadian gene expression in the peripheral clocks are not aligned to the sequence of this flow, and for this reason the SCN has been called the coordinator rather than the originator of rhythms. The phase reorganization seen in the SCN extends to the system of master-peripheral clocks. A long daylength causes phase reorganization among peripheral circadian clocks in the body, similarly to SCN subregions[5]. Feedback interactions within the SCN are expected to apply equally to the tissue-to-tissue level[6,7]. Yet, in shaping behavioral circadian rhythms the exact role played by peripheral circadian clocks is unknown.

It was recently found that the choroid plexus (CP) is also a peripheral circadian clock[8,9]. The CP, lining the third, fourth, and lateral ventricles, is one of the circumventricular organs (CVOs) that produces the cerebrospinal fluid (CSF) by actively filtering blood plasma through a monolayer of epithelial or specialized ependymal cells that surround fenestrated capillaries. The environment of the brain is closely monitored by CVOs. Found in close contact with CSF, CVOs monitor physiological parameters, such as osmolarity, and release cytokines to maintain homeostasis. The CVOs share a common structure of a tight junction-protected epithelial cell layer (blood–CSF barrier, BCSFB) enveloping the blood–brain barrier (BBB)-lacking microvessels, handling more fluid traffic than the BBB.

Separated from other body fluids, the CSF surrounds the brain and spinal cord. Tight junctions between brain endothelial cells form the BBB and keep the CSF from mixing with the blood plasma. The circulation of the CSF that floods the ventricular system, the spinal canal, and the subarachnoid cavity until resorption, is an important mechanism for brain homeostasis[10]. However, CSF homeostasis is not static and circadian variations of its production rate and composition have been demonstrated[11,12]. Production and resorption of CSF are an integral part of the glymphatic pathway that clears brain metabolites from the interstitial fluid during sleep[13,14].

Here, we looked at circadian clock activities in explant cultures from various loci of the brain and found that CVOs harbor robust circadian clocks in the brain. The presence of circadian clocks in these loci suggests that homeostatic control of the brain microenvironment is predictive and follows the circadian schedule[15], although the connection between cytokine release and transcriptional circadian oscillation is often difficult to establish. Among the CVOs, the CP stood out for its large amplitude and persistent oscillation in isolated culture. We studied the tissue-to-tissue interaction between the CP and the SCN using in vitro cocultures and reporter systems. In the presence of the CP, the long period of rhythms in the SCN is restored to the level of the

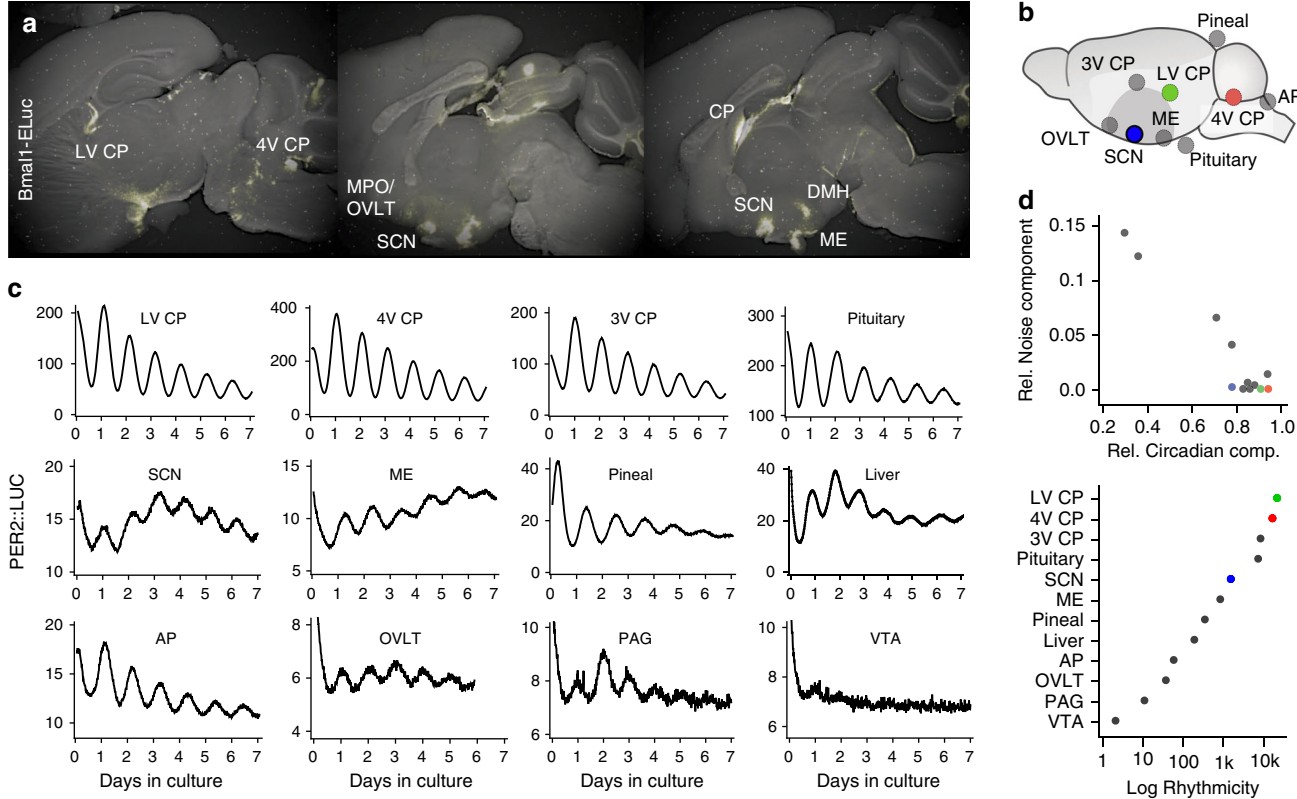

**Fig. 1** Multiple brain loci harbor circadian clocks. **a** Brain loci expressing Bmal1-ELuc. **b** Sampled brain loci for isolated explant culture. **c** Representative PER2::LUC oscillations in isolated culture, sorted by the rhythmicity score. **d** The relative circadian component and the noise component in the PER2::LUC oscillations (upper) and the rhythmicity score calculated by the ratio of the two (lower). Each dot indicates an ensemble average of tissues of the same kind (green: LV CP, red: 4V CP, and blue: SCN). The spectral power of each component is resolved by a discrete wavelet transform. LV CP choroid plexus of lateral ventricle, 4V CP choroid plexus of fourth ventricle, 3V CP choroid plexus of third ventricle, Pituitary pituitary gland, SCN suprachiasmatic nucleus, ME median eminence, Pineal pineal gland, AP area postrema, OVLT vascular organ of lamina terminalis, PAG periaqueductal gray, VTA ventral tegmental area

behavioral circadian period. We found the same trend in vivo sleep–wake cycles in mice lacking the essential circadian clock gene *Bmal1* in the CP and ependymal cells. This indicates that the CP circadian clock influences the SCN master clock and comprises an important component of the brain-wide feedback interaction of circadian clocks.

The strong oscillation of the CP is due to the high degree of synchronization of its component cellular clocks, which is surprising because a non-neuronal tissue lacks long-distance cell-to-

cell signaling despite its large geometry. We propose a simple mathematical description of the single-cell oscillation that extends to explain the unusual tissue-level synchronization of the CP clock. Amplitude and period are intertwined in oscillations in the CP, a phenomenon known as "twist". Under local coupling primarily mediated by gap junctions, synchronization can bias population-level amplitude and period, and improve spatial coherence of cellular circadian clocks. Together, our results show that the non-neuronal network of circadian clock in the CP

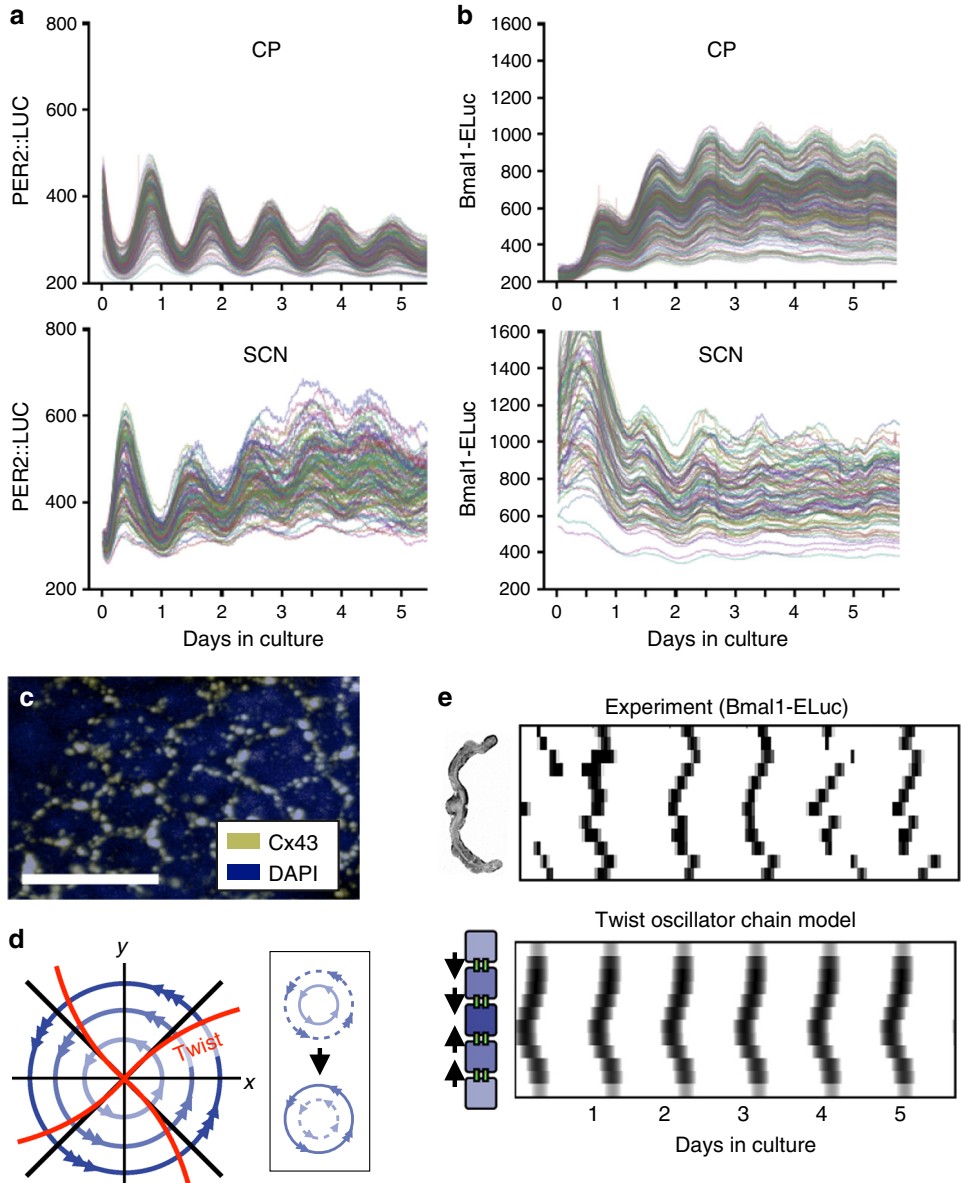

**Fig. 2** The robust circadian clock in the CP is due to strong synchronization through gap junction coupling. **a** Single-cell level circadian oscillation of PER2::LUC activity in the CP (upper) and the SCN (lower). **b** Single-cell level oscillations of Bmal1-ELuc. In the SCN, each time series tracks changes of bioluminescence in a square region-of-interest (ROI) of single-cell size. ROIs of the same dimension are used on distinct regions in the CP as single cells were visually indistinct. **c** Immunohistochemistry against the gap junction subunit Cx43 shows its expression between cells of the CP. Scale bar indicates 50 μm. **d** For self-sustained oscillators exhibiting negative twist, an increasing (instantaneous) amplitude transiently leads to a decreasing (instantaneous) period as the oscillator returns to its steady-state oscillation. Isochrons represent lines in the phase space with identical oscillator phases as time *t* goes to infinity[65] (see Methods). In systems without twist, isochrons form straight radial lines (thick black lines). In oscillators with negative twist, isochrons become bent or "twisted" (thick red line). In networks of interacting oscillators, coupling can expand the amplitude of single-cell oscillators and speed up the oscillation under negative twist (inset). See Supplementary Fig. 3 for detailed illustrations. **e** Cellular circadian oscillations in the entire 4V CP can be observed in vivo (upper left). In culture, Bmal1-ELuc oscillation shows spatial patterning of the peak phase of its circadian oscillation, which is statistically significant for the first 3–4 days (see Supplementary Fig. 4). The central region of the 4V CP phase leads (upper), which can be explained by the boundary value condition in the nearest neighbor coupling of gap junctions (lower)

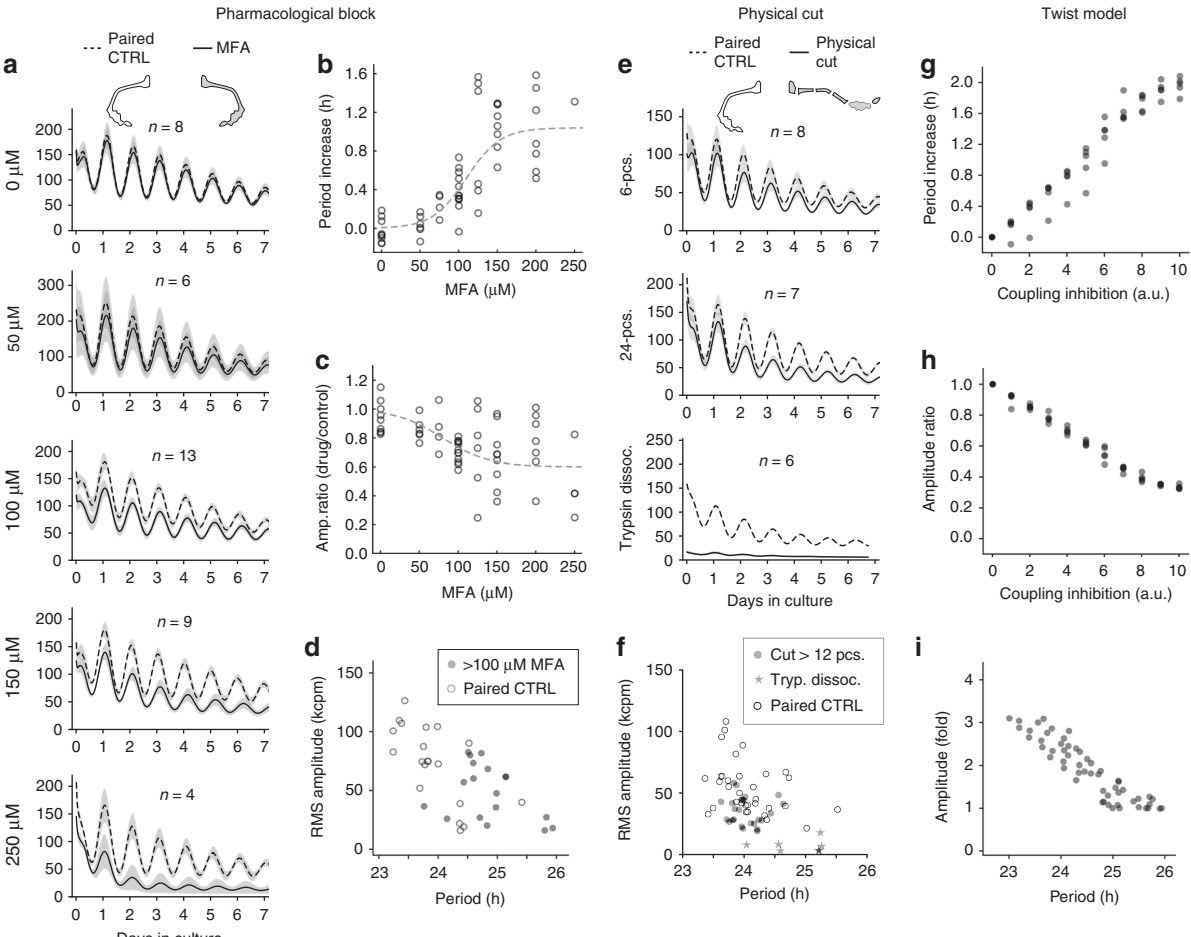

**Fig. 3** Gap junctions mediate the inverse relationship between the period and the amplitude, as the twist coupling model demonstrates. **a**–**f** The left and right pair of the 4V CP can be used as a control and a test pair to directly compare the effects of pharmacological block of gap junctions or of physical chopping along the length of the CP. **a** (From top to bottom) Shown are ensemble averages of all PER2::LUC oscillations in kcpm for the control (dashed) and the MFA-treated (solid) groups. Shades indicate the standard error of the mean. The number of samples *n* per group is indicated in each panel and the same set of data is used in **b**–**d**. **b** The period increases dose-dependently as the MFA concentration increases. Each data point is a difference between periods of the MEA-treated CP and its paired control, and indicates a data from one animal. **c** The RMS amplitude ratio of the MFA-treated CP to its paired control decreases with increasing concentration of MFA. **d** Together, an inverse relation between the period and the amplitude can be found, with the high MFA concentration group in the long-period and low-amplitude quadrant. **e**, **f** A similar trend can be found in the physically cut samples. Upper panels show the cases when the test pair of the CP is cut into 6 and 24 pieces, and when the test pair was enzyme dissociated with trypsin. **g**–**i** The twist coupling model recaptures the acceleration of the circadian oscillation and increase in amplitude by coupling. In the simulation, results from five sets of periods and initial phases randomly generated from a normal distribution (1 h standard deviation) are overlaid

makes an integral component of the brain's circadian system that provides feedback to the SCN master clock.

## Results

**Robust oscillation of the CP circadian clock.** We screened Bmal1-ELuc expression in brain loci by imaging enhanced bioluminescence from the Emerald Luciferase[16] in acute cultures of sagittal sections. The circadian clock molecule, BMAL1, is a positive regulator of the core-clock elements, PER and CRY, which form a transcriptional feedback loop at the circadian timescale. Therefore, BMAL1 can be considered the substrate for the molecular circadian clock and its expression indicates the presence of a clock. Bioluminescence reporter-based circadian clock screening has been performed previously[17,18], and the clock locations found in these studies agree with findings from systematic quantification of RNA expression[19]. Several loci were identified for putative clock hosts, including the mediobasal hypothalamus[20], areas surrounding the 3V and the brain stem[21]

(Fig. 1a; see Supplementary Fig. 1 for full scans). Putative clock loci were then dissected and cultured as isolated explants from animals expressing PER2::LUC, a bioluminescence reporter based on a fusion protein of PER2 and firefly luciferase[18]. A strong bioluminescence signal was detected in the cerebellum[22], but continued circadian rhythmicity of clock gene expression could not be confirmed in this study due to the difficulty of maintaining cerebellar tissue in culture. Weak circadian clock activities were observed in subsets of cells in the hippocampus and the cortex, but found insufficient for consistent characterization. Strong, persistent circadian expression of PER2::LUC was found mostly in the CVOs, including the CP, the pituitary gland, the median eminence (ME), the pineal gland, and the area postrema (AP), all of which are in direct contact with the CSF (Fig. 1b, c). We ranked the strength of the circadian clock by resolving the circadian oscillatory component and comparing it with the noise component in the time series (see Methods and Supplementary Fig. 2). This indicates that the strongest oscillation occurs in the

CP, followed by the pituitary gland and the SCN (Fig. 1d). The robust circadian rhythmicity of the CP stands out among all explants sampled and in all CP structures sampled from the third ventricle (3V), the lateral ventricle (LV), and the fourth ventricle (4V), despite their heterogeneous molecular identities[23]. Notably, the robustness of the CP oscillation exceeds that of the master circadian clock in the SCN.

**Tissue-wide patterning of circadian phases in the CP**. The 4V CP is particularly valuable as a model system because despite its large size, it is possible to culture and observe the whole structure. The CP can be dissected in its intact form and monitored through a cooled-CCD camera. By tracking circadian expression of PER2::LUC and Bmal1-ELuc at the single-cell level, we found that cellular circadian clocks in the CP maintain much better phase synchrony than in the SCN (Fig. 2a, b). Phases within the SCN have been shown to encode daylength and other second-order characteristics of day–night cycles[24]. Therefore, strong complete synchronization is not necessary in the neuronal network of the SCN. However, in the non-neuronal tissue network of the CP, component cellular clocks are highly synchronized. In the networks of astrocytes, endothelial cells, or endocrine cells, gap junctions mediate local coupling and create population-level dynamics, such as calcium waves[25]. Expression of gap junctions has been shown by high-throughput analysis in the CP of the mouse[26] and the rat[27]. Hence, a likely mediator of this synchronization is gap junction coupling, and we find strong expression of connexin 43 (Cx43) in the CP. Gap junctions connect neighboring cells, indicated by the hexagonal expression around cells (Fig. 2c). The CP epithelial cells, driven developmentally by FOXJ1 transcription factor, are known to express Cx43[28].

Mathematically, gap junction connections can be described by nearest neighbor coupling between single-cell circadian oscillators. Cells on the boundary end-up getting less coupling input, while cells in the central area of the tissue receive concentrated coupling input. Coupling can simultaneously affect the period and the amplitude of circadian oscillation[29] due to nonlinear interaction of transcriptional circuits[30]. A straightforward way to model the interdependence between the frequency (inverse of period) of oscillation and its amplitude is by generic oscillator models[31–33]. One of the simplest conceptual models for circadian oscillators is the modified Poincaré oscillator[34,35]. In the simplest case, the interdependence between the oscillation frequency and amplitude is assumed to be linear (see Methods). A radially outward deviation of the oscillator from its steady-state orbit causes a proportional change in period (or increase in angular velocity). This dependence relation has been referred to as "twist" or "shear" in the literature, and by convention, a negative twist parameter ($\varepsilon$) can describe an oscillator in which an amplitude increase is accompanied by a decreasing period[36] (Fig. 2d).

At the whole-tissue level, we observed spatial patterning of phases, with the leading phase appearing in the central area of the tissue (Fig. 2e, upper; Moran's $I \sim 0.5$ for initial 3–4 days, $p < 0.05$, Supplementary Fig. 4; see also Supplementary Figs. 6–8 F and G)[37]. We can explain the phase advance of the central area of the CP by the accumulation of coupling effects from the boundary (Fig. 2e, lower left). Due to resonance effects, amplitudes expand more in the central part of the chain. In a system where each oscillator has a negative twist amplitude–period relationship, the central part exhibits the shortest period. It has been previously demonstrated that short periods lead to early phases, while long periods lead to late phases upon synchronization to a mean field or an external signal[33,38]. Thus, in the fully synchronized state, the short-period central part phase advances in comparison to

lateral parts of the 4V CP. A chain of nearest neighbor-coupled Poincaré oscillators with negative twist ($\varepsilon < 0$) recaptured the observed phase relations (Fig. 2e, lower right; Supplementary Fig. 5) (see Methods for simulation details).

**The effect of twist at the population level**. The negative twist represents an inverse relationship between the period and the amplitude of an oscillator. When extended to a system of oscillators, the inverse relation strengthens as a result of an amplitude expansion as the accumulated strength of coupling grows. In the cultured 4V CP, we weakened the coupling strength by increasing concentrations of the gap junction blocker, meclofenamic acid (MFA). MFA is a water-soluble drug that potently and reversibly blocks gap junctions[39]. We took advantage of the bilaterally symmetric anatomy of 4V CP to overcome tissue-to-tissue variability. When separated into left and right sides in two dishes, both sides independently maintained perfectly identical circadian rhythms (Fig. 3a, 0 µM; $n = 9$ pairs; $p = 0.55$ for periods, paired $t$-test). We, therefore, compared PER2::LUC oscillations between the pairs. As a control, one side of the CP was cultured in a control medium, and the other side of the CP in a MFA-containing medium (Fig. 3a, top inset). This pairwise comparison was the experimental paradigm that we applied throughout (see Methods). As predicted by the twist model (Fig. 2d, e lower, and Supplementary Fig. 5), MFA dose dependently increased the period and decreased the amplitude (50–300 µM; $n = 4–13$ pairs; all data points except for outliers are indicated in Fig. 3b–d). A grid-based analysis of two-dimensional (2D) time-lapse recordings of PER2::LUC oscillations before application of MFA, during application of MFA, and after washing MFA from the medium, reveals that application of the gap junction blocker reversibly decreased global phase coherence and increased the standard deviation of the single-cell period distribution, indicating a reversible effect of the gap junction blocker (Supplementary Fig. 9). If removal of local coupling by gap junction inhibition shifts the amplitude and period along an inverse relationship, physical dissociation of the tissue is expected to cause a similar shift, albeit to a lesser degree. We cut the test side of the CP in 6–30 equal pieces, with an increment of 6. The amplitude decreased and period increased with the number of pieces, but the trend saturated after 18 pieces. The period continued to increase after enzyme dissociation, although the amplitude decrease could not be compared due to the cell loss during the dissociation process (for all physical and enzyme dissociation, $n = 5–9$ pairs; Fig. 3e, f). An analogous increase of the ensemble period can be observed in a grid-based single-cell analysis, comparing a surgically dissected 4V CP with a control (Supplementary Fig. 10). These characteristics are recaptured by the one-dimensional (1D) chain of coupled oscillators with negative twist (simulations with Gaussian random samples of periods and initial phases, $n = 4$; Fig. 3g–i). A 2D model of nearest neighbor coupled Poincaré oscillators, using a realistic geometry (see Methods for simulation details), mimics the experimentally observed phase patterns, as well as the relationship between the coupling strength, the ensemble amplitude, and the ensemble period as indicated by the above described results (Supplementary Fig. 11). It should be noted that spatial gradients of single-cell oscillator properties or coupling strength can lead also to the experimentally observed spatial patterns as discussed in the previous paragraph (Supplementary Fig. 12). Importantly, we were able to reproduce the experimentally observed dependency between coupling, amplitude, period, and phase, solely under the assumption of a negative oscillator twist (compare Fig. 3d, f with Fig. 3g–i, Supplementary Figs. 11 and 12).

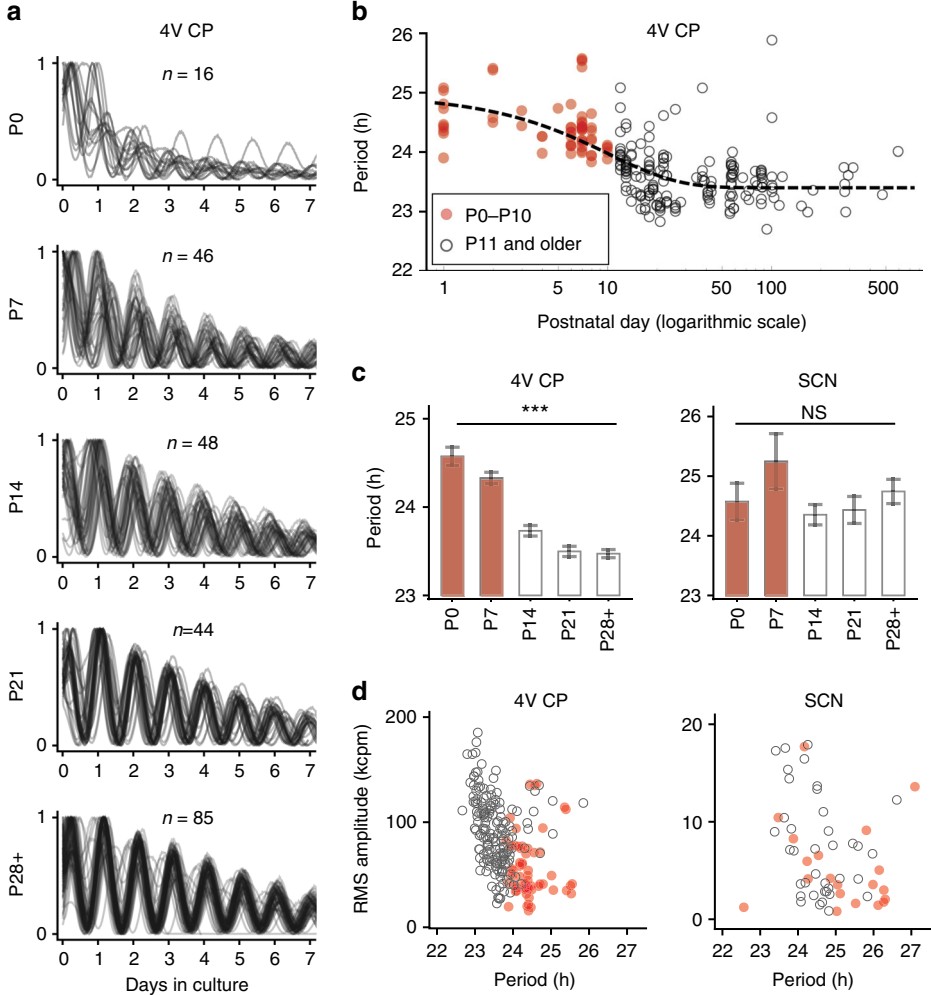

**Fig. 4** Increased synchrony and decreased period of the CP circadian clock occur during development. **a** Overlays of PER2::LUC oscillations from 4V CPs of different animals at increasing postnatal days (toward the bottom), with the initial point matched to the same hour, pooled from all control data used in this study ($n = 239$). Oscillation becomes progressively more in phase with the external light–dark cycle as age increases, and becomes more robust, indicating internal synchrony within the tissue. These developmental changes are comparable to those in the SCN in which the PER2::LUC oscillation becomes less robust past P14[40]. The number of samples (animals) is indicated in each panel. **b** Summary of the developmental change of circadian period of PER2::LUC oscillation. The dashed line indicates an exponential fit with a characteristic time of postnatal day 9.47 (period $= 1.58 \, e^{-t/9.47} + 23.41 \, \text{h}$). **c** Past the critical age between P7 and P14, the increasing speed of the CP clock stabilizes (***P0 vs P7-P28+: $p = 0.0026$, $p = 1.8 \times 10^{-7}$, $p = 2.2 \times 10^{-8}$, $p < 4.5 \times 10^{-9}$; Mann–Whitney test) which is not observed in the SCN ($n = 58$). Error bars indicate SEM. **d** In the CP, the P0–P7 group (red dots) falls into the long-period and low-amplitude quadrant, while the >P14 group (open circles) distributes in the short-period and high-amplitude quadrant along the inverse period-amplitude trend (left), while in the SCN such age dependency is absent (right). Both the CP and the SCN show an inverse relationship between the period and the amplitude, while the CP show a stronger relationship even when the sample size is controlled (CP: $R = -0.43 \pm 0.13$, $p = 5.89 \times 10^{-4}$; SCN: $R = -0.28 \pm 0.14$, $p = 0.023$). Amplitudes are rescaled by half when both left and right sides were cultured together

**Developmental change of CP period**. The inverse relationship between period and amplitude sets an operating limit for the CP circadian clock. In other words, we can predict that changing the coupling strength will alter the period of circadian gene expression oscillation in the whole tissue by simultaneously altering the amplitude. If this is true, the developmental distribution of period and amplitude in the ensemble of control samples would fall onto the inverse relationship in the same operational range (period of 23–26 h and amplitude of 20–150 kcpm or thousand photon counts per minute, Fig. 3d, f). The 4V CP is especially suitable for this kind of experiment since the entire tissue can be extracted in intact form and its size does not vary greatly compared to the changes in brain size throughout development (i.e., the CP is a relatively large structure in embryonic and neonatal brains). We

pooled data from 4V CP and SCN explants from mice at various developmental stages (postnatal days P1–P500). When we superpose PER2::LUC oscillations from ensembles of CP tissues from the same zeitgeber time, we discover that in P0, the oscillations damp quickly and the period is long, often longer than 24 h. Oscillations are not synchronized to the external light–dark cycles, as is evident in the dispersed peaks in the time series (Fig. 4a, top). The damping becomes less severe and the peaks become more coherent at P7 and the trend continues progressively as the age of the animal increases (Fig. 4a). Improvements in robustness (inverse of damping) and coherence, which often indicate stronger coupling, are associated with an ontogenetic decrease of period (Fig. 4b). The critical transition of the circadian period seems to occur at around P10 (characteristic time of

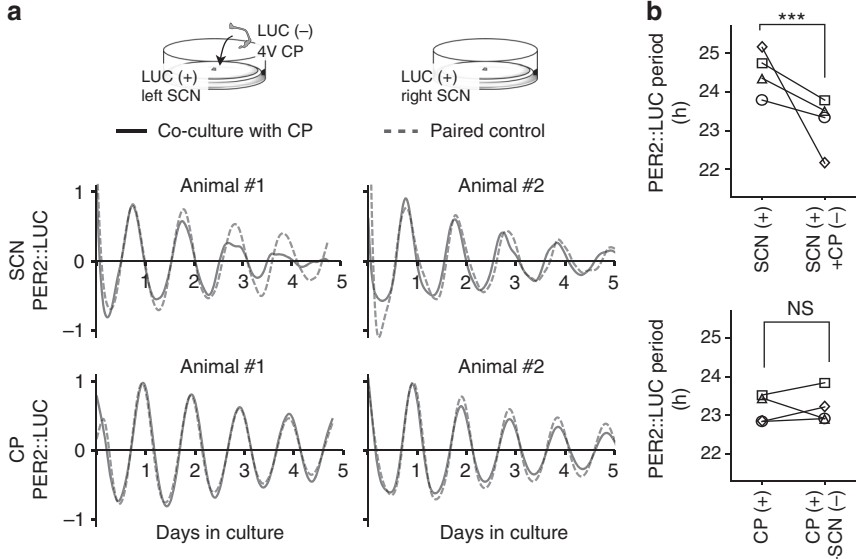

**Fig. 5** Diffusive signaling from the CP to the SCN circadian clock restores the PER2::LUC rhythm in cultured SCN close to the behavioral period. **a** When the non-bioluminescent CP is cultured directly over the SCN, the PER2::LUC oscillation in the SCN accelerates (upper), whereas the CP does not reciprocate this acceleration (lower). Two representative examples for each group are shown. The bioluminescent left and right sides of the SCN are separated to make the test (with CP) and the control (without CP) groups. **b** The coculture with the CP normalizes the PER2::LUC oscillation in the SCN, normally longer than 24.0 h in culture, to the level comparable with the behavioral period of circadian locomotor activities (23.7 h) (upper). $^{***}p < 0.001$, paired t-test. However, the CP does not accelerate its PER2:LUC oscillation when the non-bioluminescent SCN is cocultured (lower)

9.47 days as found by exponential fit). However, no such transition could be found in the SCN samples (Fig. 4c). This supports the idea that coupling in the CP strengthens with age and because of the twist in each of the component oscillators, the period-amplitude pair shifts from the lower right to upper left as an animal ages. The period-amplitude scatterplot of all control data is consistent with this hypothesis ($R = -0.43 \pm 0.06$, Pearson's correlation coefficient, $p = 1.65 \times 10^{-11}$, bootstrap z-test; Fig. 4d, left). The inverse relationship is not as steep in SCN samples ($R = -0.28 \pm 0.14$, $p = 0.023$, bootstrap z-test; Fig. 4d, right) or as significant as the CP when the sample size is controlled (CP: $R = -0.43 \pm 0.13$, $p = 5.89 \times 10^{-4}$, after size-controlled resampling). The intact SCN is anatomically less distinct and more difficult to sample than the CP and the amplitude comparison between the SCN samples may be less reliable than in the CP samples. Nonetheless, the clear twist-mediated developmental decrease of period and increase of robustness in the CP circadian clock cannot be found in the SCN, which attains robust PER2:: LUC oscillation at P7[40]. This suggests potential differences in the coupling mechanism between the two systems.

**Brain-wide coupling between the SCN and the CP**. The possibility that the CP circadian clock is flexible within an operational range leads us to speculate on its functional impact on whole-brain circadian rhythms. The SCN was long thought to be the major generator of behavioral and physiological changes in circadian rhythms of the body, until it was found that autonomous circadian clocks exist in multiple tissues[17] and cell types, including fibroblasts[41]. Although it is now thought that the SCN is a master coordinator of these clocks, we know that most of the neural connectomes contain feedback connections. We have therefore evaluated a possible interaction between the CP and the SCN in vitro. Tissue-to-tissue interaction in the culture system has been demonstrated to be feasible by placing one tissue directly above the other[42]. We once again used one lateral side of the 4V

CP and the SCN as the control, and the other side as the test group. The effect of circadian clock activity in one tissue on the other tissue can be evaluated in a pairwise manner by coculturing non-bioluminescent (LUC (−)) tissue on the bioluminescent (LUC (+)) effector tissue and comparing the bioluminescence oscillation with the contralateral partner without coculture (Fig. 5a, upper inset). The SCN under coculture with LUC (−) CP showed accelerated PER2::LUC oscillation (two representative pairs are shown in Fig. 5a, lower; complete statistics in Fig. 5b, left, $^{***}p < 0.001$, paired t-test, $n = 4$ pairs). On the other hand, no change in oscillation period was observed in the LUC (+) CP cocultured with LUC (−) SCN (Fig. 5b, right, $n = 4$ pairs). This suggests that there is an internal entrainment stream from the CP to the SCN. Interestingly, the average period of PER2::LUC oscillation in the SCN, normally longer than 24 h, became "normalized" to the behavioral circadian period of $23.7 \pm 0.1$ h[43] when cocultured with the CP (Fig. 6c). The behavioral period under constant darkness is a consistent phenotype in the C57BL/ 6J background strain ($23.83 \pm 0.04$ h in ref. [44]; PER2::LUC/ $+23.61 \pm 0.09$ h in ref. [18]).

**CP-specific silencing of the circadian clock alters behavior**. The observation that the period of PER2::LUC oscillation in cultured SCN is longer than the behavioral freerun period of locomotor activities under constant darkness[43] remains unexplained. In freely moving mice, the PER2::LUC oscillation period in the in vivo SCN matches that of the behavioral period of the animal[45]. It is therefore conceivable that the coculture condition in Fig. 5 minimally represents the normal brain environment and that the SCN circadian clock is feedback entrained through the CSF by as-yet-unidentified signaling molecules from the CP. We can experimentally test this hypothesis in vivo by selectively silencing the circadian clock in the CP. FOXJ1 is a transcription factor that induces differentiation of motile ciliated cells. In the brain, its expression is limited to the CP and ependymal cells[46].

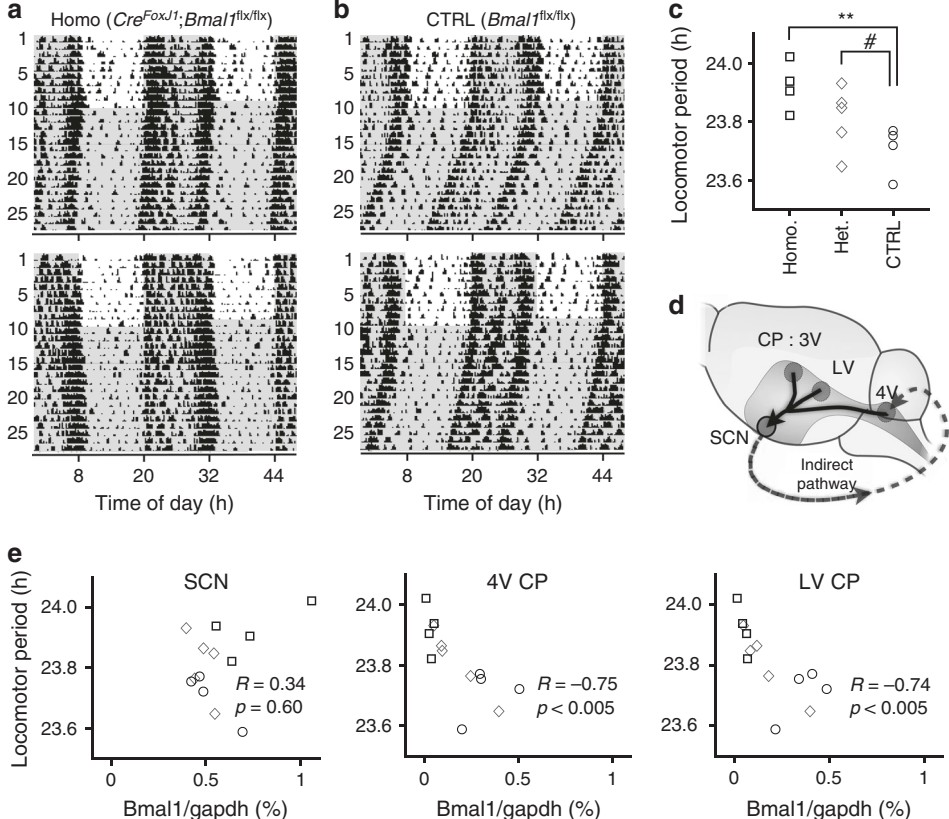

**Fig. 6 a** Targeted deletion of *Bmal1* in the CP lengthens the behavioral circadian period. Two representative double-plotted circadian actograms in mice that lack *Bmal1* in the CP. **b** Two representative double-plotted actograms in congenic controls. **c** In mice with CP-targeted *Bmal1* deletion, the circadian locomotor period becomes significantly longer than the control, which is the same as the wild type C57BL/6J (23.7 h) (homozygous knockdown (Homo., $n = 4$; open squares) against control (CTRL, $n = 4$; open circles): **$p < 0.01$; heterozygous knockdown (Het., $n = 5$; open diamonds) against control (CTRL): #$p = 0.066$, Welch's *t*-test). **d** A hypothetical diagram for connection between the CP and the SCN. The CP is likely to tune the SCN through the CSF, while the SCN relays the light signal to the CP by a non-CSF-mediated pathway. **e** In the SCN, expression of the *Bmal1* transcript is unaffected by the CP-specific Cre recombinase FOXJ1-Cre and is independent of the behavioral locomotor period. In both 4V CP and LV CP, *Bmal1* expression is close to null in the animals expressing FOXJ1-Cre. In control littermates, *Bmal1* expression varied, but was higher than zero. This makes an inverse correlation between *Bmal1* expression and the locomotor period (Pearson's correlation coefficient). Open squares, diamonds, circles indicate samples from the same animals in **c**

By crossing FOXJ1-Cre transgenic mice (*Cre^FoxJ1^*) and floxed *Bmal1* mice (*Bmal1*^flx/flx^), we generated mice with targeted knockdown of *Bmal1* in the CP (*Cre^FoxJ1^*;*Bmal1*^flx/flx^, Fig. 6a). The mean period of circadian locomotor activities under constant darkness in control *Bmal1*^flx/flx^ littermates was identical to that of the wild type (WT: 23.70 ± 0.04 h, mean ± SEM; $n = 4$, Fig. 6b). As predicted, the period increased progressively with the degree of CP-specific *Bmal1* knockdown in heterozygous mice (Het.: 23.81 ± 0.05 h, $n = 5$) and increased significantly in the homozygous knockdown (Homo.: 23.92 ± 0.04 h, $n = 4$, $p < 0.01$, Fig. 6c). This increase is consistent with the coculture results of Fig. 5.

In the brain environment, it is plausible that the CP diffuses its clock signal to the SCN through ventricular circulation of the CSF. The SCN-to-CP signaling pathway is not known, but an array of indirect pathways is feasible, such as the indirect neuronal innervation or blood-borne signaling through the neuroendocrine system (Fig. 6d). To make sure that *Bmal1* knockdown was specific to the CP, we measured *Bmal1* transcripts at the time of light-off in the same animals tested for their circadian locomotor activities. We found that the *Bmal1* transcripts were expressed similarly in the SCN across genotypes (Homo. vs WT: $p = 0.07$). However, in both the 4V CP and the

LV CP, *Bmal1* expression was close to the detection limit in *Cre^FoxJ1^*; *Bmal1*^flx/flx^ (Homo. vs WT: $p < 0.01$). We note that when samples from the heterozygous knockdown animals (*Cre^FoxJ1^*; *Bmal1*^flx/+^) were considered together, the locomotor period was highly correlated with the *Bmal1* expression level in the CP (Pearson's correlation coefficient, $R = -0.75$ (4V CP), $R = -0.74$ (LV CP), $p < 0.005$, Fig. 6e).

## Discussion

The entire brain is immersed in CSF and a large portion of the CSF is produced by the CP. Given this role, we can imagine that the CP is an important mediator of brain homeostasis. Many homeostatic processes of our bodies have a circadian timescale, which is a direct consequence of living on a planet that has a 24-h day and night cycle. The strong circadian clock in the CP can be related to the circadian homeostasis of the brain. In kidneys, the circadian clock controls circadian water homeostasis[47]. The strong CP circadian clock also contributes to the brain's circadian clock itself. Information about the day–night cycle arrives through the eyes and directly entrains circadian clock neurons in the SCN. For that reason, circadian time has been generally thought to be hierarchically organized by the SCN. However, we

discovered that the CP also influences the SCN, most likely through diffusible factors in the CSF. The CP-SCN coculture only affects the period of PER2::LUC oscillation in the SCN and not in the CP. This indicates the SCN-to-CP signaling is not mediated by the CSF, and possibly through sympathetic and parasympathetic innervation[48]. There are multiple autonomous circadian clocks in the brain, which are usually found in neuronal cells, and it is therefore surprising that the CP is composed of non-neuronal cells. There have been indications that the SCN behaves differently when isolated in vitro, potentially disputing the exclusive role of the SCN circadian clock[49,50]. The interaction between neuronal and non-neuronal circadian clock cells has recently been highlighted in the SCN, focusing on the astrocyte population[51–53]. In these reports, circadian period lengthening has been noted at both cellular and behavioral levels when the glial population loses circadian clock activities. Feedback interaction from what has been thought to be a passive receiver of circadian information may not be unusual in light of the general principle of neural organization, which heavily relies upon feedback networks. We conclude that the behavioral circadian rhythm is an integrated output of multiple clocks throughout the brain, and that the CP is an important component.

We discovered robust circadian gene expression rhythms in the CP using two independent reporters, PER2::LUC and Bmal1-ELuc. We found that circadian oscillations are spatially ordered and formulated a hypothesis of local coupling through connections by gap junctions. Modulating the degree and strength of coupling by physical cut or pharmacological gap junction inhibition weakened the amplitude of PER2::LUC oscillation, while at the same time lengthening the oscillation period. This inverse relationship between amplitude and period can be most succinctly described by a conceptual model of negative "twist" in the phase space of a circadian oscillator, in which a perturbation from the steady-state amplitude of the system accelerates the oscillation during the course of relaxation. The molecular circadian clock is a complex nonlinear system and the amplitude is an integral component intertwined with the phase. In the phase space of the PER2::LUC circadian oscillation, its orthogonal components are Per2 mRNA and PER2 protein. In our previous model, intercellular coupling drives Per2 transcription under saturable kinetics and this produces positive twist[54]. Instead, when each oscillator is driven by sequestration-based kinetics, the twist disappears[30]. Our observation of negative twist suggests that there still exists room for modeling the fundamental kinetics that drives the molecular circadian clock in mammals. In a network context, negative twist relates frequency and coupling of individual oscillators (Supplementary Figs. 12 and 13). Recent studies showed that a network of such oscillators behaves in a fundamentally different way than plain coupled oscillators[55,56]. This implies that twist is critical in how synchronization and spatial patterning arise in the CP network (see also Supplementary Discussion and Supplementary Fig. 13).

It has been noted that single-cell dispersion of the SCN widens the period distribution of PER2::LUC oscillations while also increasing the mean period[57]. Pharmacological blockade of the ubiquitous neurotransmitter in the SCN, GABA, lengthens the period of Bmal1 oscillation in equinox-entrained SCN explants[43] and in long day-entrained SCN explants[2]. Similarly, we found that inhibiting the gap junction in the CP lengthens its circadian period.

This finding raises the possibility that the circadian clock of the CP is physiologically modulated by gap junctions. Expression of the gap junction subunit Cx43 in the CP is plastic and undergoes upregulation in response to inflammation[58]. We find that prolonged treatment of gap junction blockers induces homeostatic expression of Cx43 transcript in the CP (Supplementary Fig. 14). Cx43 expression in the bladder is controlled by the circadian clock genes and coordinates urination rhythm[59]. Changes of circadian rhythms through gap junction reorganization or plasticity[60] under either physiological or pathological conditions are an important topic of future study. In our study, we demonstrated a developmental change in the CP circadian period, but its relation to gap junction expression remains unclear. Anatomical organization of the CP changes during development, but gap junction connections have not been characterized in a developmental context. CP epithelium expresses at least seven connexin subunits, including Cx43, and this complexity needs to be taken into account in future studies[61].

The nontrivial influence of the CP circadian clock on the overall behavioral rhythm offers an alternative strategy to manipulate the circadian clock using a non-SCN target. One possible avenue is pharmacological manipulation of gap junctions in the CP. It is also likely that there are other similarly important circadian clock organs in the brain and identifying them and discovering pathways that modulate their circadian oscillations could alter our view of the circadian clock as an organization of clocks.

## Methods

**Animals**. Mice under C57BL/6J background, of both sexes and ages ranging from postnatal day 0 to 600, were maintained under equinox (12 h light, 12 h dark) conditions and at constant ambient temperature. Heterozygous bioluminescence reporter-expressing strains PER2::LUC (MGI ID: 3040876)[18] and Bmal1-ELuc[16] were used for explant culture experiments. For CP-specific Bmal1 deletion, homozygous Bmal1^flx/flx and heterozygous Bmal1^flx/+; FOXJ1-Cre (MGI ID: 3797103)[46] were crossed to produce Bmal1^flx/flx littermates with the Cre (test group) or without Cre recombinase (control group). All animal protocols were reviewed and approved by the RIKEN Brain Science Institute.

**Locomotor activities**. Adult mice of different genotypes, including control littermates, were individually housed in a light–dark controlled chamber with a ventilating fan that provided constant environmental noise. Locomotor activities were monitored with a passive infrared sensor and logged on the computer as described previously[2]. Circadian periods were estimated from the chi-squared periodogram with a custom code written in Mathematica[62].

**Tissue dissection and explant culture**. Tissue culture and bioluminescence measurement protocols were identical to those used previously except for the addition of CP sampling[43]. PER2::LUC and Bmal1-ELuc mice were anesthetized and killed about 0–2 h before lights off and under dim light, brains were isolated to ice-cold HBSS within 15 s after decapitation. The brain was rinsed in fresh ice-cold HBSS and coronally cut into two close to the lambda point. For imaging-based scanning of Bmal1-ELuc expression, sagittal sections were made at 150 µm thickness on a Microslicer (DSK). For time-lapse imaging of the SCN, the anterior portion was coronally sliced at 250 µm thickness and trimmed to sample the mid-posterior section of the SCN. The posterior portion was transferred to a new dish containing ice-cold HBSS for sampling the 4V CP.

For imaging of the CP and tissue explant luminometry, select brain loci were dissected and isolated in ice-cold HBSS with forceps and a scalpel blade under the stereomicroscope. For collecting the 4V CP, the 4V was accessed by lifting the gap between the cerebellum and the brain stem, and the whole CP attached on the side of the cerebellum was gently taken out with forceps. The AP was sampled from the nearby area on the brain stem with a scalpel blade. The pineal gland was isolated from the inner side of the skull after removing surrounding dura mater and blood vessel. The whole pituitary gland was taken from the base of the skull with forceps. The whole SCN was dissected from the ventral side of the brain with a scalpel blade directly under and slightly posterior to optic chiasm. The ME and VLPO were collected along the base of 3V. Consistency of PER2::LUC activities was confirmed by comparing against tissue explants isolated from imaged slices. The anatomical location of the SCN was independently confirmed with GAD67 expression and precision of isolation was further assessed by Avp and Vip transcript expression patterns in dorsoventral subdivisions. Culture of each explant was maintained at 37 °C on the culture membrane (Millicell-CM; Millipore) in a vacuum-sealed 35-mm dish containing 1 mL of DMEM (Sigma) with NeuroBrew-21 supplement (Miltenyi Biotec), penicillin-streptomycin (25 U/mL–25 µg/mL; Nacalai Tesque), and 100 µM beetle luciferin (Promega).

**Time-lapse bioluminescence imaging and luminometry.** Single-cell resolution imaging was performed on a LV200 (Olympus) with the Orca C4742-80-12AG cooled-CCD camera (Hamamatsu) and a custom-built incubator-microscope system with the Orca R2 cooled-CCD camera (Hamamatsu). Binning was set to $4 \times 4$ and exposure time was 15 min. For whole-tissue monitoring of bioluminescence, we used a 24 dish, double photomultiplier tube (PMT) system (LM-2400; Hamamatsu) with a sampling interval of 15 min. SCN explants for whole-tissue bioluminescence were attempted in the whole SCN dissected out of the ventral side of the brain.

For gap junction inhibition, a stock solution of MFA sodium salt (Sigma-Aldrich) was prepared at ×1000 concentration in DDW and refrigerated before experiments. MFA was chosen over carbenoxolone (CBX, 400 μM) and mefloquine (MEQ, 100 μM) due to toxicity under our experimental system. The drug effect was pairwise compared in each bilateral side of the tissue (SCN and 4V CP), by culturing one side in the control medium (control) and the other side under manipulation or the drug-containing medium (test). Micro-dissection of the CP was performed on the CP mounted on the culture membrane. Enzyme dissociation of the CP was performed using TrypLE Select ×1 solution (no phenol red, Thermo Fisher Scientific) according to the manufacturer's protocol. The trypsin replacement dissociation enzyme has been conveniently called "trypsin" or "Tryp." in the main text and figures.

**Real-time quantitative PCR.** Animals from the locomotor activity monitoring chamber were anesthetized and decapitated under darkness between 1 h before and after the lights off time. The brain was isolated and immersed in ice-cold HBSS within 15 s after decapitation under dim light. Basic RNA sampling and RT-qPCR methods follow our previous protocol[2]. Whole bilateral pairs of SCN, 4V CP, and LV CP were dissected out in HBSS and each sample was transferred to 50 μl TRK lysis buffer (Omega Bio-Tek) and stored immediately at −80 °C. Total RNA was microcolumn-purified using the ENZA total RNA kit (Omega Bio-Tek) with the final elution volume of 20 μl in DEPC water. cDNA was synthesized from total RNA of 113.4 ± 18.4 ng (SCN), 318.2 ± 3.1 ng (4V CP), and 308.5 ± 5.2 ng (LV CP) (mean ± SEM) using SuperScript II RT (Thermo Fisher Scientific) with random primers. GAPDH was chosen as the internal control for each sample. RT-qPCR was performed in triplicate with SYBR Green DNA binding dye on StepOnePlus (Applied Biosystems). The following primer sequences were used (5′–3′): GAPDH forward ACGGGAAGCTCACTGGCATGG CCTT, GAPDH reverse CATGAGGTCCACCACCCTGTTGCTG; mBmal1 forward GCAGTGCCACTGACTACCAAGA, mBmal1 reverse TCCTGGA-CATTGCATTGCAT. Additional methods used for Supplementary Fig. 14 are described in Supplementary Methods.

**Immunohistochemistry.** The 4V CP sample was fixed in 4% PFA, permeabilized in 0.2% Triton X-100, blocked in 5% normal goat serum and stained slowly at low temperature (4 °C) as described in our previous work[43]. As a primary antibody, anti-connexin-43 (Cx43) rabbit polyclonal antibody (Sigma C6219, Lot. 113M4756; RRID: AB 476857) was used at dilution of 1:500–1:1000. The second antibody was anti-rabbit antibody conjugated with Alexa 488 at dilution of 1:500. The stained sample was mounted in VECTASHIELD mounting medium with DAPI (Vector Labs) before examination on an FV1000 confocal microscope (Olympus).

**Rhythmicity score.** Time series from the PMT (Fig. 1c) were analyzed by means of the multiresolution analysis (MRA) as described previously[63], using the PyWavelets software (http://pywavelets.readthedocs.io) for the Python Programming language. The MRA uses the discrete wavelet transform to decompose a given time series $s(t)$ into components (also called details) at different disjoint frequency bands[64]. In order to reduce edge effects, the first and last days of data were truncated after application of the MRA for subsequent analysis. The method has been verified on our own dataset from the SCN (Supplementary Fig. 2). Rhythmicity can be quantified by the ratio of the variance explained by the circadian component ($D_{circadian}$ or $D_6$ normalized over other components, covering from 16 h to just before 32 h) and the variance explained by the noise component ($D_{noise}$ or $D_1$ normalized over other components, covering from 0.5 h to just before 1 h). Strong circadian rhythms are associated with a large corresponding ratio. For the rhythmicity score, the components were normalized across all tissues,

$$\frac{D_{circadian}/\sum_{all\ tissues} D_{circadian}}{D_{noise}/\sum_{all\ tissues} D_{noise}} = \frac{\left(D_6^{tissues}/\sum_i D_i^{tissues}\right)/\sum_{all\ tissues}\left(D_6^{tissues}/\sum_i D_i^{tissues}\right)}{\left(D_1^{tissues}/\sum_i D_i^{tissues}\right)/\sum_{all\ tissues}\left(D_1^{tissues}/\sum_i D_i^{tissues}\right)}, \quad (1)$$

where $i = 1, 2, 3, …, 7$ indexes each of spectral component and all tissues indicate the 12 tissues as in Fig. 1d.

**Spatial autocorrelation.** Statistical significance was tested for spatial patterns in the 1D chain of Bmal1-ELuc oscillation phases, as well as for a simulated 2D grid of nearest neighbor coupled Poincaré oscillators (see Fig. 2e and Supplementary Figs. 11–13, respectively). To this end, the spatial autocorrelation index Moran's $I$ (MI) under the assumption of a nearest neighbor interaction was applied. We recently demonstrated that MI reliably detects dynamic signatures of spatial order in 2D arrays of circadian gene expression data[37]. Briefly, the MI takes into account

the cyclic nature of phase variables by

$$I := \frac{1}{\sum_{ij} w_{ij}} \frac{\sum_{ij} w_{ij} d_\theta(X_i, \overline{X}) d_\theta(X_j, \overline{X})}{N^{-1} \sum_i d_\theta(X_i, \overline{X})^2}, \quad (2)$$

where $N$ denotes the number of observables $X_i$, and $w_{ij}$ the spatial weights, and $d_\theta(X_1, X_2) := \tan^{-1}\left(\frac{\sin(X_1 - X_2)}{\cos(X_1 - X_2)}\right)$ assigns a distance between two cyclic variables $X_1$ and $X_2$. The mean value $\overline{X}$ of a set of cyclic variables $\{X_i\}_{i=1}^N$ defined by $\overline{X} := \tan^{-1}(\overline{S}/\overline{C})$ with $\overline{C} = (1/N) \sum_{i=1}^N \cos(X_i)$ and $\overline{S} := (1/N) \sum_{i=1}^N \sin(X_i)$. We also used a similar quantity that we called circular MI, $I_{circ}$, which is essentially Eq. 2 with $X_i = \exp(i\theta_i)$ and $d(x, y) = x - y$. Then,

$$I_{circ} = \frac{1}{\sum_{ij} w_{ij}} \frac{\sum_{ij} w_{ij}(e^{-i\theta_i} - \overline{X}^*)(e^{i\theta_j} - \overline{X})}{1 - \|\overline{X}\|^2}, \quad$$

where $\overline{X} = \frac{1}{N} \sum_j e^{i\theta_j}$.

Supplementary Fig. 4A shows MI values as calculated using Eq. 2 for the phase patterns in Fig. 2e (upper). Statistical significance of observed MI values was assessed by a comparison with a sampling distribution under the null hypothesis of no spatial autocorrelation, computationally determined by means of a Monte Carlo approach (see Supplementary Fig. 4B). Following a brief transitory period after transferring the tissue to the culture medium, the phase organization adopts a spatial pattern that is statistically significant for a duration of roughly 3.5 days.

**Single-cell twist model.** A modified Poincaré oscillator,

$$\frac{dr_i}{dt} = \lambda_i r_i(A_i - r_i) \quad (3a)$$

$$\frac{d\varphi_i}{dt} = \omega_i + \varepsilon_i(A_i - r_i), \quad (3b)$$

conveniently describes conceptually important oscillator properties, such as the radial relaxation rate $\lambda_i$, amplitude $A_i$, twist $\varepsilon_i$, and angular velocity $\omega_i = 2\pi/\tau_i$, where $\tau_i$ is the internal free-running period of an oscillator $i$. Eq. 3a, b describe the dynamical evolution of the radial component $r_i(t)$ and phase component $\varphi_i(t)$.

**Isochrons of the modified Poincaré model.** Isochrons are a set of initial conditions in the basin of attraction of an attracting limit cycle that have the same asymptotic phase as $t \to \infty$[65]. Importantly, isochrons provide information about the resetting and entrainment behavior of an oscillating system, such as the circadian clock. Isochrons of the oscillating system given by Eq. 3a, b can be obtained analytically.

For a given set of initial conditions $r_0$ and $\varphi_0$ at time $t_0 = 0$, the solutions of Eq. 3a, b at time $t$ can be read as

$$r(t) = \frac{A}{e^{-\lambda A t}\left(\frac{A}{r_0} - 1\right) + 1} \quad (4)$$

and

$$\varphi(t) = \varphi_0 + \omega t + \varepsilon A t + \frac{\varepsilon}{\lambda} \ln\left(\frac{A/r_0}{A/r_0 - 1 + e^{\lambda A t}}\right), \quad (5)$$

respectively. From this it follows that any dynamics starting on the limit cycle, i.e., $r_0 = A$, are given by

$$\varphi_{LC}(t) = \varphi_{0,LC} + \omega t, \quad (6)$$

and hence

$$\varphi(t) - \varphi_{LC}(t) = \varphi_0 - \varphi_{0,LC} + \frac{\varepsilon}{\lambda}\ln\left(\frac{A}{r_0}\right) - \frac{\varepsilon}{\lambda}\ln\left\{\left(\frac{A}{r_0} - 1\right)e^{-\lambda A t} + 1\right\}. \quad (7)$$

By definition, isochrons at a certain phase $\varphi_{0,LC}$ are given by all combinations of $(r_0, \varphi_0)$ that fulfill the constraint

$$\lim_{t \to \infty}\left(\varphi(t) - \varphi_{LC}(t)\right) = \varphi_0 - \varphi_{0,LC} + \frac{\varepsilon}{\lambda}\ln\left(\frac{A}{r_0}\right) \overset{!}{=} 0, \quad (8)$$

i.e., $\varphi(t)$, as well as $\varphi_{LC}(t)$ approach the same phase after the decay of transient dynamics as $t \to \infty$. Supplementary Fig. 3A–C depict examples of the vector-field, isochrons, and dependency of the oscillator period on the amplitude for different values of twist $\varepsilon \in \{−0.01, 0, 0.01\}$ h$^{-1}$ and fixed values of $A = 1, \lambda = 0.02$ h$^{-1}$, and $\tau = 24$ h, respectively.

**Coupled oscillator model and parameter constraints**. When the cells are coupled to others, coupling was introduced as an additive term on both coordinates in the Cartesian form of Eq. 3a, b as described in our previous work[35,66], using the definitions $x_i = r_i \cos(\varphi_i)$ and $y_i = r_i \sin(\varphi_i)$. Then, Eq. 3a, b become

$$\frac{dr_i}{dt} = \lambda r_i (A - r_i) + \sum_{j \neq i} K_{ij} r_j \cos\left(\varphi_j - \varphi_i\right), \quad (9a)$$

$$\frac{d\varphi_i}{dt} = \varepsilon(A - r_i) + \omega_i + \sum_{j \neq i} K_{ij} \left(\frac{r_j}{r_i}\right) \sin\left(\varphi_j - \varphi_i\right). \quad (9b)$$

If the system is sufficiently close to synchrony $\varphi_i \approx \varphi_j$ for all $(i,j)$, $\cos(\varphi_j - \varphi_i) \approx 1 - 0((\varphi_j - \varphi_i)^2)$ in Eq. 9a, and we can use mean field approximation, where $r_i$ approximately equals a neighborhood average, i.e., $\langle r_j \rangle_{j \in N_i} \approx r_i$. Then, Eq. 9a becomes

$$\frac{dr_i}{dt} \approx \lambda r_i (A - r_i) + K_i r_i, \quad (10)$$

where $K_i = \sum_{j \neq i} K_{ij}$. Then, at a stationary state ($dr_i/dt = 0$), $r_i \approx K_i/\lambda + A$ and the phase evolution is approximately

$$\frac{d\varphi}{dt} \approx \omega_i' + \sum_{j \neq i} K_{ij} \sin(\varphi_j - \varphi_i) \quad (11)$$

where $\omega_i' = \omega_i - \varepsilon K_i/\lambda$. Experimental data for amplitudes can constrain the ratio, $K_i/\lambda$, from the amplitude increase compared with the case in which gap junctions are blocked, and also $\varepsilon$ from the frequency change due to nonzero $K_{ij}$. However, $\lambda$ and $K_{ij}$ cannot be easily determined, and we chose suitable values that can replicate synchrony and MI (see above and ref. [37]) in the data (see also Supplementary Fig. 13 for how synchronization of the system and circular MI evolved when $\lambda$ and $K_{ij}$ slowly increased while $K_i/\lambda$ was held fixed in a simulation).

**1D chain model**. A 1D chain of $N$ Poincaré oscillators was constructed, each receiving coupling inputs from immediate neighbors only. Coupling was introduced as an additive term on both coordinates in the Cartesian form of Eq. 3a, b as described in our previous work[35,66], using $x_i = r_i \cos \varphi_i$, $y_i = r_i \sin \varphi_i$, and $r_i = \sqrt{x_i^2 + y_i^2}$.

$$\frac{dx_i}{dt} = (A_i - r_i)(\lambda_i x_i - \varepsilon_i y_i) - \omega_i y_i + q K_{i-1} x_{i-1} + q K_{i+1} x_{i+1} \quad (12a)$$

$$\frac{dy_i}{dt} = (A_i - r_i)(\lambda_i y_i + \varepsilon_i x_i) + \omega_i x_i + q K_{i-1} y_{i-1} + q K_{i+1} y_{i+1} \quad (12b)$$

The coupling constant $K_{ij}$ is nonzero, $K_{ij} = K_j$, only if $j = i \pm 1$ for $i = 1, 2, 3, \dots N$, while $K_0 = 0$ and $K_{N+1} = 0$. Single-cell oscillatory parameters used for the simulations in Figs. 2e and 3g–i and Supplementary Fig. 5 were $A_i = 1$, $\lambda_i = 0.02$ h$^{-1}$, and $\varepsilon_i = -0.01$ h$^{-1}$. Dose dependency of coupling inhibition was evaluated by multiples of $K_i$ with $q = 0, 1, 2, \dots 10$. In Fig. 3g–i, identical simulations were run for four random sets of Gaussian period ($\mu = 25.5$ h, $\sigma = 1$ h) and initial phase distributions ($\sigma = 1$ h).

**2D model**. In Supplementary Figs. 11, 12 and 13, the effect of twist in spatially extended systems of coupled Poincaré oscillators was analyzed for different geometries, coupling topologies and oscillator properties. In Supplementary Figs. 11 and 12, it was assumed that oscillators couple solely through a 2D nearest neighbor interaction given by a von Neumann neighborhood, i.e., the four most adjacent cells. The coupling terms for $dx_i/dt$ and $dy_i/dt$ in Eq. 12a, b were replaced with $0.5 \sum_{j=1}^{N} K_{ij} x_j$ and $0.5 \sum_{j=1}^{N} K_{ij} y_j$, respectively, where $K_{ij} = K$ whenever oscillators $i$ is influenced by oscillator $j$ through nearest neighbor interaction. In Supplementary Fig. 13, each cell coupled to neighbors within a certain distance with the same equation.

**Parameter estimation**. In the case of $N$ oscillators, we have $2N$ degrees of freedom in choosing initial conditions $x_0$ and $y_0$ and $4N$ degrees of freedom in choosing single-cell oscillator properties. In order to find a set of parameter values that qualitatively reproduced dynamical characteristics from the experimental analysis, we fixed the single-cell amplitude to $A_0 = 1$ throughout all simulations. We restricted ourselves to a nearest neighbor coupling topology as suggested by putative gap-junctional coupling and sampled the period from a normal distribution with mean 25.5 h and a standard deviation of 1 h. A mean value of 25.5 h has been chosen since period values saturate as they approach this value (Fig. 3d, f), i.e., after application of a gap junction blocker or surgical cuts. Subsequently, we determined an appropriate set of initial conditions and manually chose values for $\varepsilon$,

$\lambda$, and $K$ such that the model qualitatively reproduced the data best, i.e., the experimentally observed phase pattern is well represented and the period in case of highest coupling strength $K$ is close to the experimentally observed value of 23.5 h as indicated by Fig. 3d, f.

**Computer simulation and data analysis**. Twist chain oscillator simulations were performed on Mathematica (Wolfram Research, IL), the Python programming language using *scipy.integrate.odeint* function in SCIentific PYthon (https://scipy.org/scipylib/), and MATLAB 2016b (Mathwork, VA). Visualization and other data analyses, including periodogram and fast Fourier transform (FFT)-based period analysis, were performed on Mathematica using the custom-made *PMTAnalysis* package (http://sourceforge.net/projects/imaginganalysis)[43]. All circadian periods of the bioluminescence oscillations were estimated with FFT and all amplitudes were quantified by root mean squares of raw oscillations for equal duration of 10 recording days. Detailed methods for 2D grid-based data analysis are described in the Supplementary Methods. All statistical analyses were done on Mathematica and MATLAB 2016b by using built-in analysis functions, except for the correlation coefficients (Fig. 4d) where we computed the mean and S.E.M. by 10,000 random resampling and performed the z-test. In a size-controlled procedure for the CP data, we drew the same number of samples as the SCN data ($n = 58$).

**Data availability**. The simulation codes of illustrative examples in this study are available at modelDB (https://senselab.med.yale.edu/modeldb/) by an accession number 238338. Data analysis codes are available at https://github.com/JihwanMyung/ImagingAnalysis (*ImagingAnalysis.m* and *PMTAnalysis.m*) as Mathematica packages. Further codes and data are available on request from the corresponding authors.

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

## Acknowledgements

We thank N. Nakagawa for providing guidance on gap junction blockers, T. Suzuki and K. Yamakawa for introducing FOXJ1-Cre, J. Harrison and C. Harrison for critical reading of the manuscript, and S.D. Aird for editorial support. This work has been supported by Strategic Japanese–German Cooperative Program from the Japan Science and Technology Agency and the Deutsche Forschungsgemeinschaft (BO 3612/2-1), Grant-in-Aid for Scientific Research (16H01652, 16H06463) from the Ministry of Education, Culture, Sports, Science and Technology (MEXT), Japan, Grant-in-Aid for Scientific Research (16K08538, 15K06715, 25242077, 16H06316, 16K13110) from the Japan Society for the Promotion of Science and the Taiwan Ministry of Science and Technology Grants (104-2420-H-038-001-MY3, 106-2632-H-038-001). and The Takeda Science Foundation. J.M. acknowledges additional supports from the Wissenschaftskolleg zu Berlin and Einstein Foundation Berlin (BAK-F1-2017). C.S. acknowledges support from the Joachim Herz Stiftung. S.H. and E.D.S. are supported by OIST Graduate University. M.J.H. is supported by NIH NHLBI R01-AI130591.

## Author contributions

J.M., G.B., and T.T. designed the study. J.M. and Y.T. performed experiments. J.M., C.S., and S.H. performed analysis of experimental data. J.M., C.S., S.H., and P.R. performed

modeling. Y.Z. and M.J.H. provided the mice. J.M., C.S., S.H., M.J.H., E.D.S., and H.H. wrote the manuscript.

## Additional information

**Competing interests:** The authors declare no competing interests.

