## [Peer Review File · Nature Communications]

Reviewers' comments:

Reviewer #1 (Remarks to the Author):

The circadian clock in mammals has a hierarchical organization. Just as the suprachiasmatic nucleus (SCN) is a multi-oscillator clock, the brain contains multiple loci that maintain autonomous circadian rhythmicity. However, the role and importance of clocks other than the SCN is still an area of active research.

First, the authors examined circadian oscillations of clock gene expression in various brain loci using the PER2::*luc* and *Bmal1-Eluc* models. They discovered that the choroid plexus (CP) cells contained a robust, high-amplitude, circadian oscillation.

The choroid plexus consists of a layer of epithelial cells surrounding a core of capillaries and loose connective tissue. The CP functions as a filtration system, removing metabolic waste, foreign substances, and excess neurotransmitters from the CSF. Prior work by Nedergaard among others suggests that there are daily changes in CSF-ISF exchange caused by the expansion and contraction of the extracellular space. The authors have raised the hypothesis that the restorative properties of sleep may be linked to increased glymphatic clearance of metabolic waste products produced by neural activity in the awake brain. So the demonstration of the robust molecular rhythm in this tissue is of general interest. But the authors of the present study should do a better job in setting up the problem. The finding that there are circadian oscillators outside the SCN is hardly new. So the introduction needs to focus more on the CP itself.

Fig. 1 shows pictures of *Bmal1-Eluc* expression in the CP but rhythms appear to come from the PER2::*luc* mice. This inconsistency should be fixed.

The authors go on to use computational analysis and modeling show that the CP achieves these properties by synchronization of 'twist' circadian oscillators via gap junctional connections. While we liked the inclusion of modeling data, we needed a little more explanation about the model and found this part hard to evaluate. Also more justification for exploring the role of gap junctions would be helpful.

The authors do show that single cells from the CP express circadian rhythms in PER2::*Luc* and that a gap junction blocker disrupts the population rhythm. We would like to see single cell phase, period, amplitude mapping of CP. For all of this data, we would like to know the sample size as it was unclear. Also, are single cells impacted by the gap junction blocker or just the population rhythms? What cell types are expressing these rhythms? Finally, can the authors confirm that the gap junction blocker is indeed blocking gap junctions and not just killing the CP cells?

We are glad to see that developmental data although this experiment is not well integrated with the rest of the work.

Finally, the authors speculate that the CP clock tunes the SCN clock through circulation of

cerebrospinal fluid and adjusts behavioral circadian rhythms. This is an important and novel hypothesis. Experimentally, they show that the co-culture of CP and SCN can decrease SCN PER2::*LUC* period. We would like to see longer rhythms as 4-5 days is standard rather than the 2 days that are shown. In general, the first 24 hour of PER2::*LUC* in tissue culture are unstable and it is very difficult to accurately measure the period. We would also like to see an increased sample size and ideally some attempt to define the coupling agent. Even some basic biochemistry with heat-inactivation or molecular size exclusion would be helpful here.

Finally, the authors use a targeted *Bmal1* KO to eliminate the molecular block from the CP and show that this impacts behavior. Of course, *FOXJ1-cre* is not CP specific which should be discussed by the authors, still this is the beginning of a really exciting finding. Here it would be important to demonstrate that the CP-tissue from the transgenic mouse lacks the *Per2>::luc* rhythm but, more importantly, fails to alter the SCN rhythm of WT tissue. This experiment would help address the question of whether it is the molecular clock in the CP tissue that is influencing the SCN rhythm or just the cells altering the media through some non-rhythm factor.

So, in summary, some of the findings are really interesting and novel. The possible finding that the CP could influence the SCN is very exciting. However, more experimental work is needed to convince the reader. Furthermore, the experiments described in this manuscript are not well integrated nor is the story clearly told.

Reviewer #2 (Remarks to the Author):

Summary: In this manuscript, the authors characterize a new mammalian circadian multi-oscillator. It is in the choroid plexus and apparently influences the rhythms of the well-studied "master" circadian clock in the suprachiasmatic nucleus. The experimental evidence is clear and the results collectively tell a consistent story. The authors use a phenomenological mathematical model to add new insights to the data, making it easy for the reader to form a mental model of the newly discovered clock. Further, this manuscript is well-written and mathematical modeling is integrated tightly with the experimental results.

Highlight: The authors have not only found an important secondary "clock", they have used a mathematical description that is precise, yet allows the reader to develop an intuition for what makes these oscillators similar (they are circadian oscillators that adjust their phases and amplitudes in response to intercellular signals) but different (as the oscillators gain amplitude, they also speed up). Does this amplitude growth/speed-up combination arise from a particular type of interaction in the underlying gene regulatory network? Previous work has shown which kinetics create the opposite effect and which create a neutral effect. It will be interesting to see how this new effect drives further study of a mechanistic understanding of the clock.

Concern: Why use a 1D chain to model a 2D grid? Given that the influence of each neighbor is an important part of the interpretation, it would make the connection between the model

and the data stronger if the topology were more similar. The difference between a 1D and a 2D topology could be significant.

Recommendation: The work in this manuscript identifies a new region of the brain that has demonstrated effects on the master clock. This makes it high-impact and, in my opinion, suitable for publication in Nature Communications. I strongly recommend it for publication, assuming my above concern is addressed. Not only is the content high-impact, but the presentation is concise and clear. The figures are effective and the figure captions give a short sentence indicating what we should conclude from the figure and then enough detail to understand how to read the figure and come to one's own conclusion. The text is clear and each paragraph in the results refers to one figure. The effect is that the reader can comprehend a lot of information quickly.

Typographical errors:

Introduction paragraph 1. Final sentence. "in shaping behavioral circadian rhythm" should be "in shaping behavioral circadian rhythms"

Results "robust oscillations" section. Final sentence. "that of the master clock,SCN" should be "that of the master clock in the SCN"

Results "tissue wide patterning" section. "the accumulation of the coupling effect from the boundary" should be "the accumulation of the coupling effects from the boundary"

Reference and citations. Instead of "John PCS and Doyle FJ", it should be "St John PC and Doyle III FJ". The citation should be to (St John and Doyle, 2015) or (St John and Doyle III, 2015). I realize that different journals have different specific requirements. The important point is that "St. John" is the first author's last name.

Figure 2 Caption. In part D, "When the oscillator, isochron, in the phase space" is awkward and I am not sure how to fix it.

Figure 3 Caption. In part (G-I). "results from five sets of periods and phases" should be "results from five sets of periods and initial phases"

Reviewer #3 (Remarks to the Author):

The authors present an integrated experimental and modeling study to investigate the role of the Choroid Plexus (CP) in circadian rhythm generation. This work is very novel as previous research has focused on circadian oscillators in the Suprachiasmatic Nucleus (SCN) or external tissues with little regard to the possibility of oscillators in other brain regions. The authors present a convincing set of experiments that support the idea that the CP is a non-neuronal circadian oscillator and that the CP oscillator could reduce the period of the SCN. I believe that these results are important contributions to the circadian field.

The authors nicely integrate modeling studies to interpret the experimental data. A simple phenomenological model based on modified Poincare oscillators is used. While I found the modeling results to be interesting, a disadvantage of the using this simple model is that few biophysical insights can be extracted. The idea that the system exhibits negative twist can be extracted from the data alone without the use of a model. Perhaps the authors could better emphasize what novel conclusions result from the modeling exercise. It also would be useful if the authors could explain how the model parameters were tuned and how negative twist might be predicted with a biophysically-based model.

Overall this paper represents an impressively executed and important study on the role of the Choroid Plexus (CP) in circadian rhythm generation.

Reviewer #4 (Remarks to the Author):

Myung et al. describe how the choroid plexus (CP) maintains a circadian clock and how it interacts with the SCN. The combination of modeling and experiments convincingly shows that the CP can sustain its own circadian clock, most likely relying gap junctions, and interact with the SCN to decrease the SCN period, which matches the observed periodicity of the activity of mice. In addition, the authors show how the CP gains synchrony over the first postnatal days of mice. Computational modeling relying on a simple oscillator model recapitulates the experimental observations and strengthen the message.

In general, I found that the manuscript well written and easy to understand. It addresses an important question about the role and independence of the different circadian clocks found in the brain. The results are overall convincing although I have comments described below that should be addressed. I think that the presented research is of value to be considered for publication in Nature Communication but I would strongly suggest to clarify multiple important points about the cell-cell interaction and improve a few key technical and statistical aspects.

Major points about cell-cell interaction, synchronization and modeling

1. The authors claim that the 'most likely mediator of the [CP] synchronization is gap junction' (page 3). The authors show rather convincingly that the synchronization is due to direct interaction in fig. 3 but is it a known fact? What are the alternative hypotheses? Does the statement rely on the known literature or does it precede the results presented by the authors?

2. In addition, the authors mention that 'the CP also entrains the SCN, most likely through diffusible factors in the CSF' (discussion). Would it be possible that the diffusible factors have a role in how the CP is synchronized? Results in fig. 3 suggest that not, but the authors should make this clearer. In addition, it may be interesting to test the role of a global synchronizing agent in the computational model to either rule out its role, or show that it is not sufficient (i.e. local interactions are required to maintain synchrony).

3. Related to 2, how realistic is it to consider the CP as a 1-d object. How would the results change if the model is 2-dimensional with the same local interaction, or when including a global synchronizing agent. In addition, the model presented in fig. 2E have only 7 cells; how would the results hold if the model comprises a number of cells closer to the actual size of the CP? How does 7 cells compare to the size of the CP presented in the experimental data in fig. 2E, top right?

4. There is published work on the type of network architectures found in the SCN as well as across and between different circadian oscillators. Is the network architecture important for the CP too? E.g. would a scale-free network show a phase shift between the center and edges? Regarding the interaction between the CP and SCN, it may also be interesting to hypothesize how diffusible factors may relate to the architecture of the SCN (would it act on all SCN neurons; only the core entrained by light, or only the shell?)

5. Given that the CP gains synchrony during the first 10 days of life in mice, are there known differences in protein expression. In particular, is the gap junction density increasing up to day 10? Or are proteins involved in junctions, like connexin 43, more expressed after day 10?

Technical points to address

6. In page 4: 'A chain of nearest neighbor-coupled Poincaré oscillators with negative twist ($e < 0$) recaptured the observed phase relations' (also in legend of fig. 2E). What is the statistic used for this claim? Related to the point 2, can other modeling hypotheses be compared.

7. The interpretation of fig. 4D is misleading. The significance of the correlation is only driven by the larger number of data points in the CP plot. The SCN plot has higher (yet still low) absolute correlation value, but it is less significant because of fewer data points. I understand that data collection is hard but I would not overstate the interpretation of this particular result.

8. How were the parameters chosen in the model ('Twist modeling' in the method section)? And how sensitive are the results to small variations of these parameters? A brief explanation of the choice and/or a small sensitivity analysis would be beneficial. In addition, the authors mention that 'identical simulations were run...', but do not describe the outcome of these simulations (e.g. how the variability compare to the difference with the experimental data). The quantification of this comparison relates to the point 6 raised above.

9. What is the underlying equation in the fit for fig. 3B, C and fig. 4B. Is there is a reason for choosing a sigmoidal curve? Is there something additional that can be learnt from the fit parameters?

Minor points

10. There is missing information about the number of animals used and how many replicates have been done. Sentences like (page 4): 'the period and decreased the amplitude (50-300 μ M; n = 4-13 pairs; all data points except for outliers are indicated in Fig. 3B-D)' and legend of fig. 2E, fig. 4A, and fig. 4C should be clarified even in cases where the sample number can be count in the figure.

11. Page 4: '(period of 23-26 h and amplitude 20-150 kcpm or thousands of photon counts per minute, Fig. 3D & 3F).' I would suggest to use either kcpm or 'thousands of photon counts per minute'.

12. The test used in fig. 4C is not mentioned. In addition, the author could perform the test with different groupings to show convincingly that 10 days is the cutoff with the highest significance.

13. Page 5: '(two representative pairs are shown in Fig. 5A, lower; complete statistics in Fig. 5B, left, $p < 0.001$, n = 4 pairs)'. Which test was used?

14. Page 5: '...became 'normalized' to the behavioral circadian period of 23.7 h of the C57BL/6 background strain...'. The actual result of 23.7 h comes later in the text. It is confusing given that it is written as a fact without a reference. The authors should either cite a reference or reorganize the flow to present it as a result of their own even if the explanation comes later.

15. The wording around eq. 3 is confusing because x and y are not explicitly defined. I guess that these the values are the phase-space for a one-dimensional model, but it may be confused for the amplitude in a two-dimensional model. I would suggest to clarify it.

16. Microscopy images lack a scale bar.

17. Legend of fig. 3 is very sparse. In particular, the y-axis of 3B and 3C are not well defined.

18. X-axis scales for panels in fig. 4D are not the same. Enforcing the same x range for these two plots would help the comparison.

19. Legend of fig. S2 (8th and 9th panels) is on top of the plot.

We thank anonymous reviewers for their thoughtful comments that led to this revision (NCOMMS-17-18402A) entitled, "The choroid plexus as an important circadian clock component." We have carefully considered the comments and tried to improve the presentation. Here are our detailed responses.

Detailed Response to Reviewers

Reviewer #1 (Remarks to the Author):

The circadian clock in mammals has a hierarchical organization. Just as the suprachiasmatic nucleus (SCN) is a multi-oscillator clock, the brain contains multiple loci that maintain autonomous circadian rhythmicity. However, the role and importance of clocks other than the SCN is still an area of active research.

*First, the authors examined circadian oscillations of clock gene expression in various brain loci using the PER2::*luc* and Bmal1-Eluc models. They discovered that the choroid plexus (CP) cells contained a robust, high-amplitude, circadian oscillation.*

The choroid plexus consists of a layer of epithelial cells surrounding a core of capillaries and loose connective tissue. The CP functions as a filtration system, removing metabolic waste, foreign substances, and excess neurotransmitters from the CSF. Prior work by Nedergaard among others suggests that there are daily changes in CSF-ISF exchange caused by the expansion and contraction of the extracellular space. The authors have raised the hypothesis that the restorative properties of sleep may be linked to increased glymphatic clearance of metabolic waste products produced by neural activity in the awake brain. So the demonstration of the robust molecular rhythm in this tissue is of general interest. But the authors of the present study should do a better job in setting up the problem. The finding that there are circadian oscillators outside the SCN is hardly new. So the introduction needs to focus more on the CP itself.

Answer: We found this to be an important advice. We took time to carefully revise the introduction to give more focus on the choroid plexus in light of circadian rhythmicity. We appreciate the reviewer for this comment.

*Fig. 1 shows pictures of Bmal1-Eluc expression in the CP but rhythms appear to come from the PER2::*luc* mice. This inconsistency should be fixed.*

Answer: We tried to address this concern. In Fig. 1, Bmal1-ELuc was used for a screening purpose and clock expression was checked in detail using PER2::LUC. The scanning of potential clock loci in a wide view-field (and low numerical aperture) was possible because of the bright bioluminescence of Emerald Luciferase (ELuc) used in Bmal1-ELuc. We attempted the same with PER2::LUC which uses conventional firefly luciferase but could not attain sufficient quality to determine small clock loci except for the choroid plexus. Bmal1 can be considered a substrate for the circadian transcriptional circuit and its existence can be a good indicator of circadian clock's presence, which was confirmed with PER2::LUC in explants of isolated brain loci. In the text, we specified these technical details.

The authors go on to use computational analysis and modeling show that the CP achieves these properties by synchronization of 'twist' circadian oscillators via gap junctional connections. While we liked the inclusion of modeling data, we needed a little more explanation about the model and found this part hard to evaluate. Also more justification for exploring the role of gap junctions would be helpful.

Answer: We expanded explanation on the mathematical model in the main text, the Materials and Methods section as well as in the caption of Figure 2D. We described the single cell oscillator model, the 1D chain model and the 2D model in further detail and specified the parameter estimation process.

We have focused on gap junctions because of desynchronizing effects by different kinds of gap junction blockers. We narrowed our focus on connexin 43 (Cx43) because among 20 connexin family members in mice, it is the most commonly expressed subtype found in epithelial cells (Goodenough & Paul, 2009). Gap junctions are known mediators of cell-to-cell communication that have been previously shown to be expressed in mouse and rat choroid plexus (Marques *et al.*, 2011; Liddelow *et al.*, 2013). In urinary bladder epithelium, Cx43 confers circadian rhythmicity to urinary bladder capacity (Negoro *et al.*, 2012). Choroid plexus on its surface expresses ciliated epithelial cells, which are developmentally driven by FOXJ1 transcription factor, and express Cx43 (Huang *et al.*, 2003). Finally, we found evidence for homeostatic upregulation of Cx43, and not Cx36, expression after its prolonged blockade by connexin blockers 18-beta-glycyrrhetic acid (bGA) or halothane, and not TTX, in culture. This is shown in the new Supplementary Figure S14.

The authors do show that single cells from the CP express circadian rhythms in PER2::Luc and that a gap junction blocker disrupts the population rhythm. We would like to see single cell phase, period, amplitude mapping of CP. For all of this data, we would like to know the sample size as it was unclear. Also, are single cells impacted by the gap junction blocker or just the population rhythms? What cell types are expressing these rhythms? Finally, can the authors confirm that the gap junction blocker is indeed blocking gap junctions and not just killing the CP cells?

Answer: We analyzed two-dimensional bioluminescence recordings of *in vitro* preparations of the choroid plexus. Since single cells of the CP are generally not identifiable with a high enough accuracy in order to track them throughout the experiment, a pixel based approach has been chosen. The tissue throughout the analyzed experiments appeared to be stable enough to justify this decision. A summary of the two-dimensional phase, period and amplitude together with the pixel-based raw time series and a “skeleton-analysis” corresponding to Figure 2E of the main text can be seen in the new Supplementary Figures S6-S8. Generally, a high phase coherence, as well as a stable period and amplitude can be observed (an exception is Figure S7 where a lower signal to noise ratio due to the camera setup can be observed). In some cases, a spatial patterning of phase with a leading central part, similar to the findings in Figure S2E of the main text, can be observed. Interestingly, in these cases the amplitude shows a significant positive correlation between the phase difference from the mean phase and the amplitude, see corresponding panels H.

Finally, we analyzed in Figure S9 two-dimensional bioluminescence recordings before, during and after the application of MFA. A reversible decline of the global phase coherence as well as a reversible broadening of the period distribution could be observed during the application of MFA. The recovery of the global phase coherence after washing MFA from the medium indicates that blocking gap junctions does not kill CP cells at least under the investigated concentration of 200 micro-molar MFA.

We are glad to see that developmental data although this experiment is not well integrated with the rest of the work.

Answer: We tried to highlight that the developmental changes falls under the general inverse relation between amplitude with more detailed description on statistics.

Finally, the authors speculate that the CP clock tunes the SCN clock through circulation of cerebrospinal fluid and adjusts behavioral circadian rhythms. This is an important and novel hypothesis. Experimentally, they show that the co-culture of CP and SCN can decrease SCN

PER2::LUC period. We would like to see longer rhythms as 4-5 days is standard rather than the 2 days that are shown. In general, the first 24 hour of PER2::LUC in tissue culture are unstable and it is very difficult to accurately measure the period. We would also like to see an increased sample size and ideally some attempt to define the coupling agent. Even some basic biochemistry with heat-inactivation or molecular size exclusion would be helpful here.

Answer: We have replaced the representative traces in Fig. 5A for recording duration of 5 days. Identification of the coupling agent is an important question. We tried two approaches, application of a putative agent vasopressin and application of sampled CSF to the cultured SCN, but could not find a conclusive answer within our resources and time. Thus, identifying the coupling agent would have to be left for a new future project.

Finally, the authors use a targeted Bmal1 KO to eliminate the molecular block from the CP and show that this impacts behavior. Of course, FOXJ1-cre is not CP specific which should be discussed by the authors, still this is the beginning of a really exciting finding. Here it would be important to demonstrate that the CP-tissue from the transgenic mouse lacks the Per2::luc rhythm but, more importantly, fails to alter the SCN rhythm of WT tissue. This experiment would help address the question of the whether it is the molecular clock in the CP tissue that is influencing the SCN rhythm or just the cells altering the media through some non-rhythm factor.

So, in summary, some of the finding are really interesting and novel. The possible finding that the CP could influence the SCN is very exciting. However, more experimental work is needed to convince the reader. Furthermore, the experiments described in this manuscript are not well integrated nor is the story clearly told.

Answer: We are grateful to the reviewer for their excitement and support. In the revision, we tried to highlight our arguments while also discussing weaknesses, such as details of expression of FOXJ1-Cre. Due to relocation of the main author (J.M.), PER2::LUC rhythm could not be checked in the triple mutant FOXJ1-Cre-driven Bmal1 flox PER2::LUC strain. However, we trust this will not be a serious concern for misinterpretation. We directly verified absence of Bmal1 expression in the CP of this strain, and in Bmal1 KO, peripheral tissues fail to oscillate PER2::LUC (Ko, Yamada, *et al*, *PLoS Biol*, 2010). It is possible that Bmal2 replaces the role of Bmal1 but in mice, Bmal1 controls expression of Bmal2 so knocking down Bmal1 has an effect of Bmal1/Bmal2 double knock-down (Shi *et al*, *Curr Biol*, 2010). Co-culture of rhythmic and non-rhythmic pair of the CP and the SCN is an interesting experiment, yet, we would have to leave this for a next project. We thank once again for detailed comments to integrate the story, which helped the manuscript improve greatly.

Reviewer #2 (Remarks to the Author):

Summary: In this manuscript, the authors characterize a new mammalian circadian multi-oscillator. It is in the choroid plexus and apparently influences the rhythms of the well-studied "master" circadian clock in the suprachiasmatic nucleus. The experimental evidence is clear and the results collectively tell a consistent story. The authors use a phenomenological mathematical model to add new insights to the data, making it easy for the reader to form a mental model of the newly discovered clock. Further, this manuscript is well-written and mathematical modeling is integrated tightly with the experimental results.

Highlight: The authors have not only found an important secondary "clock", they have used a mathematical description that is precise, yet allows the reader to develop an intuition for what makes these oscillators similar (they are circadian oscillators that adjust their phases and

amplitudes in response to intercellular signals) but different (as the oscillators gain amplitude, they also speed up). Does this amplitude growth/speed-up combination arise from a particular type of interaction in the underlying gene regulatory network? Previous work has shown which kinetics create the opposite effect and which create a neutral effect. It will be interesting to see how this new effect drives further study of a mechanistic understanding of the clock.

Concern: Why use a 1D chain to model a 2D grid? Given that the influence of each neighbor is an important part of the interpretation, it would make the connection between the model and the data stronger if the topology were more similar. The difference between a 1D and a 2D topology could be significant.

Answer: One-dimensional approximations of higher-dimensional systems have previously been successfully applied, e.g., to describe the electrical activity of the small intestine (Keener & Sneyd, 2009). However, the reviewer is right that a two-dimensional topology can lead to qualitative changes of the dynamics. Thus, we investigated the temporal evolution in a two-dimensional array of Poincaré oscillators for different geometries, oscillator properties and coupling topologies. The new Supplementary Figure S11 shows that indeed also a two-dimensional array of nearest-neighbor-coupled Poincaré oscillators is able to reproduce the experimentally observed phase-patterns. It should be noted that generally the long-term solutions of coupled spatially-extended systems is strongly dependent on the initial conditions and oscillator properties. The coupling-period-relationship as experimentally observed (Figure 3B of the main text) could only be recovered with the Poincaré oscillator exhibiting negative twist, see new Supplementary Figure S11D and E. Finally, it should be noted that any potential spatial gradient in oscillator or coupling topology can robustly lead to a specific pattern. A higher coupling strength in the central or lateral parts of the CP, will lead to higher amplitude expansion in these regions and, thus, to a shorter period (in case of negative twist) which ultimately leads to an earlier phase, see Supplementary Figure S12.

Recommendation: The work in this manuscript identifies a new region of the brain that has demonstrated effects on the master clock. This makes it high-impact and, in my opinion, suitable for publication in Nature Communications. I strongly recommend it for publication, assuming my above concern is addressed. Not only is the content high-impact, but the presentation is concise and clear. The figures are effective and the figure captions give a short sentence indicating what we should conclude from the figure and then enough detail to understand how to read the figure and come to one's own conclusion. The text is clear and each paragraph in the results refers to one figure. The effect is that the reader can comprehend a lot of information quickly.

Typographical errors:

Introduction paragraph 1. Final sentence. "in shaping behavioral circadian rhythm" should be "in shaping behavioral circadian rhythms"

Results "robust oscillations" section. Final sentence. "that of the master clock,SCN" should be "that of the master clock in the SCN"

Results "tissue wide patterning" section. "the accumulation of the coupling effect from the boundary" should be "the accumulation of the coupling effects from the boundary"

Reference and citations. Instead of "John PCS and Doyle FJ", it should be "St John PC and Doyle III FJ". The citation should be to (St John and Doyle, 2015) or (St John and Doyle III, 2015). I realize that different journals have different specific requirements. The important point is that "St. John" is the first author's last name.

Figure 2 Caption. In part D, "When the oscillator, isochron, in the phase space" is awkward and I am not sure how to fix it.

Figure 3 Caption. In part (G-I). "results from five sets of periods and phases" should be "results from five sets of periods and initial phases"

Answer: We thank the reviewer and modified the text accordingly.

Reviewer #3 (Remarks to the Author):

The authors present an integrated experimental and modeling study to investigate the role of the Choroid Plexus (CP) in circadian rhythm generation. This work is very novel as previous research has focused on circadian oscillators in the Suprachiasmatic Nucleus (SCN) or external tissues with little regard to the possibility of oscillators in other brain regions. The authors present a convincing set of experiments that support the idea that the CP is a non-neuronal circadian oscillator and that the CP oscillator could reduce the period of the SCN. I believe that these results are important contributions to the circadian field.

The authors nicely integrate modeling studies to interpret the experimental data. A simple phenomenological model based on modified Poincare oscillators is used. While I found the modeling results to be interesting, a disadvantage of the using this simple model is that few biophysical insights can be extracted. The idea that the system exhibits negative twist can be extracted from the data alone without the use of a model. Perhaps the authors could better emphasize what novel conclusions result from the modeling exercise. It also would be useful if the authors could explain how the model parameters were tuned and how negative twist might be predicted with a biophysically-based model.

Overall this paper represents an impressively executed and important study on the role of the Choroid Plexus (CP) in circadian rhythm generation.

Answer: We are glad that the reviewer appreciates the results and importance of our study. Regarding the choice to study phenomenological oscillator models: Such phenomenological or conceptual oscillator models found wide applications in studies of circadian systems (Roenneberg *et al.*, 2008; Gonze, 2011). Due to the simplicity of such models, useful lessons can be learned about how fundamental oscillator properties such as the amplitude, period, relaxation rate, or twist of an oscillator affect its PRC, entrainment to Zeitgeber cycles (Oda and Otto Friesen, 2011; Schmal *et al.*, 2015), or synchronization to other oscillatory entities (Abraham *et al.*, 2010). Since these intrinsic oscillator properties are implemented in a straight-forward manner and not hidden behind the complexity of multi-dimensional biophysical models (gene-regulatory networks and networks of gene-regulatory networks), the interpretation of the results is tremendously facilitated. Importantly, lessons learned from these conceptual models can later be transferred to better understand the behavior of more complicated contextual biophysical models.

We modified the main text in order to better introduce and justify conceptual oscillator models and appended the References along these lines. Interestingly, we found twist also in biophysically motivated oscillator models. Even the modified Goodwin oscillator which can be considered the prototype of a biochemical negative feedback oscillator shows twist, i.e., an alteration of its (instantaneous) period after the perturbation from its steady state amplitude (data not shown). Additionally, parameter variations that simultaneously lead to alterations of period and amplitude can be interpreted as twist and one could refer to this aspect as "parametric twist". It is plausible that external processes such as synchronization and entrainment can act on specific parameters,

showing such twist effects. Our analysis might help to interpret these situations. Such simultaneous variations of periods and amplitudes upon parameter variations are common in molecular circadian oscillator models, see e.g., the sensitivity analysis in Spreadsheet S1 of (Korencic *et al.*, 2012).

Reviewer #4 (Remarks to the Author):

Myung et al. describe how the choroid plexus (CP) maintains a circadian clock and how it interacts with the SCN. The combination of modeling and experiments convincingly shows that the CP can sustain its own circadian clock, most likely relying gap junctions, and interact with the SCN to decrease the SCN period, which matches the observed periodicity of the activity of mice. In addition, the authors show how the CP gains synchrony over the first postnatal days of mice. Computational modeling relying on a simple oscillator model recapitulates the experimental observations and strengthen the message.

In general, I found that the manuscript well written and easy to understand. It addresses an important question about the role and independence of the different circadian clocks found in the brain. The results are overall convincing although I have comments described below that should be addressed. I think that the presented research is of value to be considered for publication in Nature Communication but I would strongly suggest to clarify multiple important points about the cell-cell interaction and improve a few key technical and statistical aspects.

Major points about cell-cell interaction, synchronization and modeling

1. The authors claim that the ‘most likely mediator of the [CP] synchronization is gap junction’ (page 3). The authors show rather convincingly that the synchronization is due to direct interaction in fig. 3 but is it a known fact? What are the alternative hypotheses? Does the statement rely on the known literature or does it precede the results presented by the authors?

Answer: The answer relates to the answer on the third question of Reviewer 1. In summary, it is known from the literature that gap junctions are mediators of cell-to-cell communication (Benninger *et al.*, 2008). Cx43 is the most commonly expressed subtype among all 20 connexin family members in epithelial cells (Goodenough & Paul, 2009), and have been specifically found to be expressed in rat and mice CPs (Marques *et al.*, 2011; Liddelov *et al.*, 2013). In new Supplementary Figure S9, we show that the application of the gap junction blocker MFA leads to a reversible decline in the global phase coherence and a broadening of the period distribution. It is plausible to assume that these effects occur due to weakened inter-cellular coupling upon MFA application. Finally, we also found evidence for homeostatic upregulation of Cx43, and not Cx36, expression after its prolonged blockade by connexin blockers bGA2 or halothane, and not TTX (Figure S14).

2. In addition, the authors mention that ‘the CP also entrains the SCN, most likely through diffusible factors in the CSF’ (discussion). Would it be possible that the diffusible factors have a role in how the CP is synchronized? Results in fig. 3 suggest that not, but the authors should make this clearer. In addition, it may be interesting to test the role of a global synchronizing agent in the computational model to either rule out its role, or show that it is not sufficient (i.e. local interactions are required to maintain synchrony).

Answer: We agree that diffusible factors generally might play an important role in maintaining synchrony among oscillatory units. However, we found oscillatory patterns of clock gene reporter expression in the CP that are significantly different from random patterns, see Figures 2E and the new Supplementary Figures S6-S9. Such patterns of, e.g., oscillatory phases, cannot be produced by a global or mean-field synchronization agent under the assumption of identical or randomly drawn

single cell oscillator properties as we have shown in a previous analysis (Schmal *et al.*, 2017a). However, another possibility is the assumption of spatial gradients in single cell oscillator properties. In this scenario even mean-field coupling can lead to spatial patterning. The impact of spatial gradients in coupling strength is shown in the new Supplementary Figure S12. We shortly discussed these issues in the revised main text.

3. Related to 2, how realistic is it to consider the CP as a 1-d object. How would the results change if the model is 2-dimensional with the same local interaction, or when including a global synchronizing agent. In addition, the model presented in fig. 2E have only 7 cells; how would the results hold if the model comprises a number of cells closer to the actual size of the CP? How does 7 cells compare to the size of the CP presented in the experimental data in fig. 2E, top right?

Answer: We thank the reviewer for this suggestion. In the revised manuscript, we discuss common features and differences between a one dimensional-chain of coupled oscillators and a two-dimensional spatially-extended system of coupled oscillators. Indeed, also a global synchronizing agent can reproduce the period-amplitude relationship under the assumption of twist. However, under mean-field coupling, the observed phase patterning can solely be reproduced under the assumption of an additional gradient or specific pattern of single cell oscillator properties, e.g., a specific spatial distribution of single cell periods or amplitudes. We would also like to refer to the answer given to Reviewer 1, regarding a deeper discussion of the two-dimensional model.

Furthermore, we thank the reviewer for pointing out that we indeed separated the CP into 12 regions of interest for further analysis in Fig2 E (top) while simulating a linear chain of only 7 coupled oscillators. We made new simulations based on 12 oscillators and modified Fig. 2E (bottom).

4. There is published work on the type of network architectures found in the SCN as well as across and between different circadian oscillators. Is the network architecture important for the CP too? E.g. would a scale-free network show a phase shift between the center and edges? Regarding the interaction between the CP and SCN, it may also be interesting to hypothesize how diffusible factors may relate to the architecture of the SCN (would it act on all SCN neurons; only the core entrained by light, or only the shell?)

Answer: The answer partially relates to our answer to Reviewer 2 and point number 2. In order to generate patterns that are significantly different from random patterns in spatially extended systems of coupled oscillators, one needs either local interactions as shown in (Schmal *et al.*, 2017a) or spatial gradients of single cell oscillator properties and/or coupling schemes (see above answers). Along these lines, it is plausible that under specific conditions also a scale-free network might lead to a phase shift between the center and the edges. Scale-free networks are characterized by hubs, i.e., the in-degree distribution of nodes follows a power law such that some nodes (hubs) receive more input (coupling) than others. Thus, if a hub is placed in the center of the CP, the higher coupling strength received by the hub might lead to a higher amplitude expansion which in turn leads to a lower period (in case of negative twist) and ultimately to an earlier phase in this region. These considerations are closely related to the studies of the new Supplementary Figure S12, where a spatial gradient in coupling strength has been assumed that subsequently led to early phases in regions of higher coupling.

Heterogeneity in the network coupling can be very important, particularly when correlation between the frequency (phase) and coupling strength is present as in our twisted oscillator model. Recent studies showed that such an oscillator network can behave quite differently from traditional models, particularly when its structure is scale-free. The network shows so-called “explosive synchronization”, meaning that, as coupling strength slowly increases from zero, an initial desynchronized state turns into a synchronized state via the first-order phase transition, unlike the second-order one in a (untwisted) Kuramoto model (Gómez-Gardeñes *et al.*, 2011; Zhang *et al.*,

2013). Although these studies investigated networks with much more extensive connectivity than ours, but assuming that a similar principle applies, we can suggest that synchronization in the CP network arises in the process of nucleation and growth, just as other examples with the first-order phase transition (e.g, boiling water). In this case, oscillators with stronger couplings, either via spatial heterogeneity in coupling strength (e.g, Fig. S11, S12) and/or connectivity structure (e.g., scale-free networks), can play important roles in generating and maintaining global oscillation as loci that initiate and spread synchrony. In a new Supplementary Figure S13, we simulated and compared an untwisted and twisted 2D oscillator network where coupling strength gradually increased from zero. Here, twisted oscillators reached good global synchrony later than untwisted ones, whereas Moran's I , a measure of spatial correlation, became larger in the twisted than untwisted case (Fig. S13A). This indicates that, synchronized oscillation in the twisted case begins with appearance of locally synchronized oscillators, which we can call nucleation of synchrony. Indeed, when we compared the states of two networks at the similar synchronization level, we can see a nucleation-like blob of oscillators that are locally synchronized together in the twisted case, particularly around the center where oscillators are more connected (Fig. S13B). On the other hand, in the untwisted case, such oscillators are scattered all around the network.

We also note that this aspect makes the CP different from previously studied oscillator networks: To our extent of knowledge, biological studies that used coupled oscillator models have almost never considered twist. For example, a beautiful study by Benninger *et al.* (2008) on Ca^{2+} wave propagation in the pancreatic islet demonstrated how the gap junction coupling enables the synchronized oscillation and its propagation, which are phenomenologically similar to the CP. However, behavior of this model is similar to percolation, which arises via the second-order transition. Therefore, the CP is potentially a system that can operate under currently unknown and novel principles, but this is certainly beyond the scope of our study and we want to explore this in future studies. In a revised manuscript, we discussed these issues in Discussion and Supplementary Discussion along with Fig. S13.

Regarding the CP-SCN interaction, we have not collected data suitable for the questions that the reviewer raised. However, we can still discuss the implications based on our finding that the CP effectively pushes the SCN oscillators forward. We (JM, SH, EDS, and TT) previously showed that the similar "push-forward" interaction is also present between the dorsal (~core) and ventral (~shell) part of the SCN and is critical for seasonal time coding by the SCN network. Therefore, if the CP influences only the dorsal or ventral SCN oscillators, it can undermine the SCN network coding, unless there are homeostatic mechanisms compensating for this. Therefore, on this theoretical ground, we can suggest that it is more likely that the CP affects the whole SCN in the same way. However, given the lack of data to support or disprove this, we leave this question for future studies.

5. Given that the CP gains synchrony during the first 10 days of life in mice, are they known differences in protein expression. In particular, is the gap junction density increasing up to day 10? Or are proteins involved in junctions, like connexin 43, more expressed after day 10?

Answer: This is a difficult question to answer and we are afraid that experimentally exploring this is beyond the scope of our current resources and time. We speculate this is likely (as in the homeostasis study shown in the answer to reviewer#1's question) but the protein-level activation is not guaranteed by the transcriptional expression. Liddelow *et al.* shows continued expression of Cx43 gap junction in adults but at this stage, we are unable to answer this question.

Technical points to address

6. In page 4: 'A chain of nearest neighbor-coupled Poincaré oscillators with negative twist ($e < 0$) recaptured the observed phase relations' (also in legend of fig. 2E). What is the statistic used for this claim? Related to the point 2, can other modeling hypotheses be compared.

Answer: Regarding the first part of the question, we added a paragraph to the Materials and Methods section of the main text in order to explain our parameter estimation procedure in more detail. In summary, parameters have been manually chosen such that key features of the experimental data are qualitatively reproduced, e.g., a certain phase pattern in combination with the experimentally observed coupling-period-amplitude relationship, see Figure below [A) Simulated time series B) Comparison of mean-centered phase at time point $t = 4$ d in Fig. 3E (top) of main text and the mean-centered phase at time $t = 4$ d in the simulation of panel A)].

In regard to point 2, it should be noted that we have enough degrees of freedom in the mathematical model in order to reproduce the phase-patterns as observed in experiments. For example, as stated above one can always construct gradients of oscillator properties and/or coupling strengths such that the system exhibits a certain phase pattern. Importantly, the negative twist hypothesis additionally explains the relationship between coupling, amplitude and period for different coupling strength as found in experiments with pharmacological treatments or surgical cuts of the CP, see e.g., Fig. 3 and Fig.4.

7. The interpretation of fig. 4D is misleading. The significance of the correlation is only driven by the larger number of data points in the CP plot. The SCN plot has higher (yet still low) absolute correlation value, but it is less significant because of fewer data points. I understand that data collection is hard but I would not overstate the interpretation of this particular result.

Answer: In the revised manuscript, we provide a better statistical comparison based on a new bootstrap procedure with data size control. We computed the mean and standard deviation for correlation coefficients from 10,000 repetitions randomly resampled samples from each data set, and performed the z-test with a mean and variance from the bootstrap samples. The size control is made by resampling only $n=58$ samples from the CP data that match the size of the SCN data. This showed that the negative relationship between the amplitude and period were present both in the CP and SCN while the CP had a stronger tendency. This information is provided in the main text and Fig. 4D, and Materials and Methods.

8. How were the parameters chosen in the model ('Twist modeling' in the method section)? And how sensitive are the results to small variations of these parameters? A brief explanation of the choice and/or a small sensitivity analysis would be beneficial. In addition, the authors mention that 'identical simulations were run...', but do not describe the outcome of these simulations (e.g. how the variability compare to the difference with the experimental data). The quantification of this comparison relates to the point 6 raised above.

Answer: Our answer partially relates to the answer given to point 6 of this reviewer. In this revision, we explained the parameter estimation process and clarified the modeling in the substantially extended Materials and Methods section. In summary, parameters have been manually chosen such that key features of the experimental data are qualitatively reproduced, e.g., a certain phase pattern in combination with the experimentally observed coupling-period-amplitude relationship, see for example the new Supplementary Figure S11. Figure S 11 D-E shows the impact of different coupling strength K on the ensemble amplitude and period of the system for a previously chosen

radial relaxation rate and twist parameter. It should be noted, that the behavior for different K is sensitive to single cell oscillator properties such as twist or the relaxation rate as discussed in (Schmal *et al.*, 2017b). For example, a larger radial relaxation rate would lead to less amplitude expansion and thus to a reduced shortening of the period, assuming an un-altered value for the twist parameter.

9. *What is the underlying equation in the fit for fig. 3B, C and fig. 4B. Is there is a reason for choosing a sigmoidal curve? Is there something additional that can be learnt from the fit parameters?*

Answer: We have used a sigmoidal curve for fitting but used it primarily as a guide for the eye. The parameters we obtained set the range of parameters we were used for simulations.

Minor points

10. *There is missing information about the number of animals used and how many replicates have been done. Sentences like (page 4): ‘the period and decreased the amplitude (50-300 μ M; n = 4-13 pairs; all data points except for outliers are indicated in Fig. 3B-D)’ and legend of fig. 2E, fig. 4A, and fig. 4C should be clarified even in cases where the sample number can be count in the figure.*

Answer: One data point (difference or ratio of control-test pair) is from one animal. We have indicated this in corresponding figure legends, specifically in Figure 3 and Figure 4.

11. *Page 4: ‘(period of 23-26 h and amplitude 20-150 kcpm or thousands of photon counts per minute, Fig. 3D & 3F).’ I would suggest to use either kcpm or ‘thousands of photon counts per minute’.*

Answer: Since the cpm is a generally used term, we kept kcpm while adding thousand photon counts per minute.

12. *The test used in fig. 4C is not mentioned. In addition, the author could perform the test with different groupings to show convincingly that 10 days is the cutoff with the highest significance.*

Answer: We performed Mann-Whitney test for different groupings and testing all combinations up to P14 group showed statistical significance against P21 ($p=0.007$) and P28+ ($p=0.00003$). We have performed a simple exponential fit of the raw data in Fig 4B and found the characteristic time constant to be 9.47 postnatal days. In signal processing, this is accepted as the cutoff timescale. The details are provided in the corresponding legend.

13. *Page 5: ‘(two representative pairs are shown in Fig. 5A, lower; complete statistics in Fig. 5B, left, $p < 0.001$, n = 4 pairs)’. Which test was used?*

Answer: The paired Student's *t*-test was used. We have specified this in the revision.

14. *Page 5: ‘...became ‘normalized’ to the behavioral circadian period of 23.7 h of the C57BL/6 background strain...’. The actual result of 23.7 h comes later in the text. It is confusing given that it is written as a fact without a reference. The authors should either cite a reference or reorganize the flow to present it as a result of their own even if the explanation comes later.*

Answer: In the revised manuscript, we refer to both literature data (Myung *et al.*, 2012; Hofstetter *et al.*, 1995; Yoo *et al.*, 2003) and Figure 6C.

15. The wording around eq. 3 is confusing because x and y are not explicitly defined. I guess that these the values are the phase-space for a one-dimensional model, but it may be confused for the amplitude in a two-dimensional model. I would suggest to clarify it.

Answer: We thank the reviewer for pointing to the missing definitions of x and y which we included in the revised manuscript.

16. Microscopy images lack a scale bar.

Answer: We added a scale bar in the image.

17. Legend of fig. 3 is very sparse. In particular, the y-axis of 3B and 3C are not well defined.

Answer: We expanded the legend of Fig. 3 and explained each panel. Also, we fixed a wrong annotation in Fig 3D and F (kcps to kcpm).

18. X-axis scales for panels in fig. 4D are not the same. Enforcing the same x range for these two plots would help the comparison.

Answer: We re-plotted Fig. 4D and the CP and the SCN panels have the same x-axis scale. We have also replaced the solid, half-transparent gray dots with solid, half-transparent red dots to improve visibility.

19. Legend of fig. S2 (8th and 9th panels) is on top of the plot.

Answer: We modified the caption accordingly.

References (Letter)

Abraham U, Granada AE, Westermarck PO, Heine M *et al.* (2010). Coupling governs entrainment range of circadian clocks. *Mol Syst Biol.* **6**: 438.

Benninger RK, Zhang M, Head WS, Satin LS *et al.* (2008) Gap junction coupling and calcium waves in the pancreatic islet. *Biophys J.* **95**: 5048-6

Goodenough DA & Paul DL (2009). Gap junctions. *Cold Spring Harb Perspect Biol.* **1**: a002576.

Gonze D (2011). Modeling circadian clocks: from equations to oscillations. *Cent Eur J Biol.* **6**: 699-711.

Gómez-Gardeñes J, Gómez S, Arenas A, & Moreno Y (2011) Explosive synchronization transitions in scale-free networks. *Phys Rev Lett.* **106**: 128701

Hofstetter JR, Mayeda AR, Possidente B, & Nurnberger JI Jr. (1995). Quantitative trait loci (QTL) for circadian rhythms of locomotor activity in mice. *Behav Genet.* **25**: 545-56.

Huang T, You Y, Spoor MS, Richer EJ, *et al.* (2003). FoxJ1 is required for apical localization of ezrin in airway epithelial cells. *J Cell Sci.* **116**: 4935-4945.

Keener J and Sneyd J (2009). Chapter 18.3.3 in *Mathematical Physiology II: Systems Physiology*, 2nd Ed., Antman SS, Marsden JE, & Sirovich L eds. New York, NY: Springer, 8/II.

- Ko CH, Yamada YR, Welsh DK, Buhr ED, *et al.* (2010). Emergence of noise-induced oscillations in the central circadian pacemaker. *PLoS Biol.* **8**: 10.
- Korencic A, Bordyugov G, Kosir R, Rozman D, *et al.* (2012). The interplay of *cis*-regulatory elements rules circadian rhythms in mouse liver. *PLOS ONE.* **7**: e46835.
- Liddelow SA, Dziegielewska KM, Ek CJ, Habgood MD, *et al.* (2013). Mechanisms that determine the internal environment of the developing brain: a transcriptomic, functional and ultrastructural approach. *PLOS ONE.* **11**: e0147680.
- Marques F, Sousa JC, Coppola G, Gao F, *et al.* (2011). Transcriptome signature of the adult mouse choroid plexus. *Fluids Barriers CNS.* **8**: 10.
- Myung J, Hong S, Hatanaka F, Nakajima Y, *et al.* (2012). Period coding of *Bmal1* oscillators in the suprachiasmatic nucleus. *J Neurosci.* **32**: 8900-8918.
- Negoro H, Kanematsu A, Doi M, Suadicani SO, *et al.* (2012). Involvement of urinary bladder Connexin 43 and the circadian clock in coordination of diurnal micturition rhythm. *Nat Commun.* **3**: 809.
- Oda GA & Otto Friesen W (2011). Modeling two-oscillator circadian systems entrained by environmental cycles. *PLoS ONE* **6**: e23895.
- Pett JP, Korencic A, Wesener F, Kramer A, & Herzog H (2016). Feedback Loops of the Mammalian Circadian Clock Constitute Repressilator. *PLOS Comp Biol.* **12**: e1005266.
- Roenneberg T, Chua EJ, Bernardo R, & Mendoza E (2008). Modelling biological rhythms. *Curr Biol.* **18**: 826-835.
- Schmal C, Myung J, Herzog H, & Bordyugov G (2015). A theoretical study on seasonality. *Front Neurol.* **6**: 94.
- Schmal C, Myung J, Herzog H, & Bordyugov G (2017a). Moran's *I* quantifies spatio-temporal pattern formation in neural imaging data. *Bioinformatics.* **33**: 3072-3079.
- Shi S, Hida A, McGuinness OP, Wasserman DH, *et al.* (2010). Circadian clock gene *Bmal1* is not essential; functional replacement with its paralog, *Bmal2*. *Curr. Biol.* **20**: 316-321.
- Yoo SH, Yamazaki S, Lowrey PL, Shimomura K, *et al.* (2003). PERIOD2::LUCIFERASE real-time reporting of circadian dynamics reveals persistent circadian oscillations in mouse peripheral tissues. *Proc Natl Acad Sci USA.* **101**: 5339-5346.
- Zhang X, Hu X, Kurths J, & Liu Z (2013) Explosive synchronization in a general complex network. *Phys Rev E.* **88**: 010802.

REVIEWERS' COMMENTS:

Reviewer #1 (Remarks to the Author):

The authors have done a good job addressing most of our concerns. The findings are really interesting and novel. The experiments and the modeling work are better described in the revised version.

From our stand point, a key missing experiment is the demonstration that the CP-tissue from the transgenic mouse lacks the *Per2::luc* rhythm but, more importantly, fails to alter the SCN rhythm of WT tissue. This experiment would help address the question of the whether it is the molecular clock in the CP tissue that is influencing the SCN rhythm or just the cells altering the media through some non-rhythm factor.

The authors argue that they are not in a position to do this experiment and have done a lot of nice work. Therefore, in balance, we would recommend this work for publication in Nature Communications.

Reviewer #2 (Remarks to the Author):

The authors have fully addressed my concerns and I strongly support publication of this manuscript.

Reviewer #3 (Remarks to the Author):

I am satisfied with the authors' modifications to the description of the modeling work based on my first review.

Reviewer #4 (Remarks to the Author):

I congratulate the authors for the extensive revisions and improved manuscript. I found the corrections useful and mostly addressing my concerns as well as the ones from the other reviewers. I thus recommend the manuscript for publication in Nature Communications.

Detailed Response to Reviewers (NCOMMS-17-18402B)

Reviewer #1 (Remarks to the Author):

The authors have done a good job addressing most of our concerns. The findings are really interesting and novel. The experiments and the modeling work are better described in the revised version.

*From our stand point, a key missing experiment is the demonstration that the CP-tissue from the transgenic mouse lacks the *Per2::luc* rhythm but, more importantly, fails to alter the SCN rhythm of WT tissue. This experiment would help address the question of the whether it is the molecular clock in the CP tissue that is influencing the SCN rhythm or just the cells altering the media through some non-rhythm factor.*

The authors argue that they are not in a position to do this experiment and have done a lot of nice work. Therefore, in balance, we would recommend this work for publication in Nature Communications.

Answer: We appreciate the reviewer for many valuable comments. We would look forward to follow-up experiments that clarify some of the remaining unknowns.

Reviewer #2 (Remarks to the Author):

The authors have fully addressed my concerns and I strongly support publication of this manuscript.

Answer: Thank you for the strong support.

Reviewer #3 (Remarks to the Author):

I am satisfied with the authors' modifications to the description of the modeling work based on my first review.

Answer: Thank you for improving our model with helpful comments.

Reviewer #4 (Remarks to the Author):

I congratulate the authors for the extensive revisions and improved manuscript. I found the corrections useful and mostly addressing my concerns as well as the ones from the other reviewers. I thus recommend the manuscript for publication in Nature Communications.

Answer: We are grateful for your review.